# Mice with endogenous TDP-43 mutations exhibit gain of splicing function and characteristics of amyotrophic lateral sclerosis

Pietro Fratta[1,†,‡,*] , Prasanth Sivakumar[1,‡], Jack Humphrey[1,2,‡], Kitty Lo[2,‡], Thomas Ricketts[3,‡], Hugo Oliveira[3,‡], Jose M Brito-Armas[4], Bernadett Kalmar[1], Agnieszka Ule[1], Yichao Yu[5], Nicol Birsa[1], Cristian Bodo[1], Toby Collins[1], Alexander E Conicella[6], Alan Mejia Maza[1], Alessandro Marrero-Gagliardi[4], Michelle Stewart[7], Joffrey Mianne[7] , Silvia Corrochano[3], Warren Emmett[2], Gemma Codner[7], Michael Groves[1], Ryutaro Fukumura[8], Yoichi Gondo[8], Mark Lythgoe[5], Erwin Pauws[9], Emma Peskett[9], Philip Stanier[9] , Lydia Teboul[7], Martina Hallegger[10], Andrea Calvo[11], Adriano Chiò[11], Adrian M Isaacs[1,12] , Nicolas L Fawzi[13] , Eric Wang[14], David E Housman[14], Francisco Baralle[15], Linda Greensmith[1], Emanuele Buratti[15] , Vincent Plagnol[2], Elizabeth MC Fisher[1,†] & Abraham Acevedo-Arozena[3,4,†,**]

## Abstract

TDP-43 (encoded by the gene *TARDBP*) is an RNA binding protein central to the pathogenesis of amyotrophic lateral sclerosis (ALS). However, how *TARDBP* mutations trigger pathogenesis remains unknown. Here, we use novel mouse mutants carrying point mutations in endogenous *Tardbp* to dissect TDP-43 function at physiological levels both *in vitro* and *in vivo*. Interestingly, we find that mutations within the C-terminal domain of TDP-43 lead to a gain of splicing function. Using two different strains, we are able to separate TDP-43 loss- and gain-of-function effects. TDP-43 gain-of-function effects in these mice reveal a novel category of splicing events controlled by TDP-43, referred to as "skiptic" exons, in which skipping of constitutive exons causes changes in gene expression. *In vivo*, this gain-of-function mutation in endogenous *Tardbp* causes an adult-onset neuromuscular phenotype accompanied by motor neuron loss and neurodegenerative changes. Furthermore, we have validated the splicing gain-of-function and skiptic exons in ALS patient-derived cells. Our findings provide a novel pathogenic mechanism and highlight how TDP-43 gain of function and loss of function affect RNA processing differently, suggesting they may act at different disease stages.

**Keywords** ALS; cryptic exon; skiptic exon; splicing; TDP-43
**Subject Categories** Molecular Biology of Disease; Neuroscience; RNA Biology
**The EMBO Journal (2018) 37: e98684**

See also: **C Rouaux** *et al* (June 2018)

1   UCL Institute of Neurology, and MRC Centre for Neuromuscular Disease, London, UK
2   UCL Genetics Institute, London, UK
3   MRC Mammalian Genetics Unit, Harwell, UK
4   Unidad de Investigación, Hospital Universitario de Canarias, Fundación Canaria de Investigación Sanitaria and Instituto de Tecnologías Biomédicas (CIBICAN), La Laguna, Spain
5   UCL Centre for Advanced Biomedical Imaging, University College London, London, UK
6   Graduate Program in Molecular Biology, Cell Biology and Biochemistry, Brown University, Providence, RI, USA
7   MRC Mary Lyon Centre, Harwell, UK
8   Mutagenesis and Genomics Team, RIKEN BioResource Center, Tsukuba, Ibaraki, Japan
9   UCL Institute of Child Health, London, UK
10  UCL Institute of Neurology and Francis Crick Institute, London, UK
11  Rita Levi Montalcini Department of Neuroscience, University of Turin, Turin, Italy
12  UK Dementia Research Institute at UCL, UCL Institute of Neurology, London, UK
13  Department of Molecular Pharmacology, Physiology & Biotechnology, Brown University, Providence, RI, USA
14  Department of Biology, Massachusetts Institute of Technology, Cambridge, MA, USA
15  International Center for Genomic Engineering and Biotechnology (ICGEB), Trieste, Italy
    *Corresponding author. Tel: +44 2034 484112; E-mail: p.fratta@ucl.ac.uk
    **Corresponding author. Tel: +34 9226 78108; E-mail: aacevedo@ull.edu.es
    †These authors contributed equally to this work
    ‡These authors contributed equally to this work

## Introduction

TDP-43 is a ubiquitously expressed, predominantly nuclear, RNA binding protein, which is involved in multiple steps of RNA processing and maturation, including transcription and splicing (Ratti & Buratti, 2016; Ederle & Dormann, 2017). TDP-43 has two RNA binding domains (RRM1 and RRM2) and a C-terminal low complexity glycine-rich domain (LCD), in which mutations causative for amyotrophic lateral sclerosis (ALS) are clustered (Ratti & Buratti, 2016; Ederle & Dormann, 2017).

Mutations in *TARDBP* account for a small percentage (< 5%) of ALS cases; however, cytoplasmic TDP-43 inclusions, accompanied by nuclear depletion of the protein, are the key pathological hallmark of > 98% of ALS indicating the profound importance of TDP-43 in pathogenesis (Mackenzie *et al*, 2010; Sreedharan *et al*, 2008); furthermore, TDP-43 deposition has been consistently described in a range of other neurodegenerative diseases most prominently frontotemporal dementia (Neumann *et al*, 2006).

Despite this, the mechanism(s) by which TDP-43 mutations lead to disease remains unknown, and both nuclear loss of function (LOF) and cytoplasmic gain of function (GOF) have been proposed to play a role (Ling *et al*, 2013). TDP-43 deposition and nuclear depletion in post-mortem tissue support a potential role for LOF in end-stage ALS. However, little is known about the early-stage effects of *TARDBP* mutations, including the impact on RNA metabolism (Koyama *et al*, 2016).

TDP-43 is extremely dosage sensitive and is tightly autoregulated by a mechanism that involves TDP-43 protein binding to its own 3′UTR; these properties have proven challenging for studying the effects of *TARDBP* mutations *in vivo* (Ayala *et al*, 2011; Polymenidou *et al*, 2011; Avendaño-Vázquez *et al*, 2012). Thus, the main approach to studying LOF, by using null mice, has been hampered because TDP-43 knockout (KO) causes very early embryonic lethality, and heterozygous mice have normal levels of TDP-43 due to autoregulation (Kraemer *et al*, 2010; Sephton *et al*, 2010; Wu *et al*, 2010; Ricketts *et al*, 2014). However, conditional KO strategies and downregulation of TDP-43 via siRNA suggest that *acute* TDP-43 loss of function could lead to neurodegeneration (Chiang *et al*, 2010; Wu *et al*, 2012; Iguchi *et al*, 2013; Yang *et al*, 2014).

Similarly, in transgenic mice, even low levels of overexpression of wild-type (WT) or mutant TDP-43 cause multiple RNA changes, making it difficult to identify the pathogenic RNA profile that occurs physiologically in disease (Arnold *et al*, 2013).

Here, we have addressed these issues by working with animals from an allelic series carrying point mutations within mouse endogenous *Tardbp* to dissect the molecular effects of TDP-43 LOF and GOF *in vivo* in a physiological setting and in the absence of the confounding effects of ectopically expressed transgenes. One of our novel mouse mutants has decreased TDP-43 RNA binding capacity, which allowed us to characterise TDP-43 LOF. A second strain carries a mutation within the C-terminal low complexity glycine-rich domain (LCD) of TDP-43, which induces a splicing GOF. The analysis of this GOF uncovered entirely novel splicing events in genes not known to be controlled by TDP-43, such that a set of constitutive exons—here named "skiptic exons" (SE)—are skipped due to TDP-43 GOF.

Remarkably, the *skipping* of constitutive exons induced by GOF is in contrast to the previously identified TDP-43 LOF-induced cryptic exons, which are normally repressed but in LOF are aberrantly *included* in mRNAs (Ling *et al*, 2015; Humphrey *et al*, 2017).

Furthermore, and importantly for modelling of TDP-43 disease processes, our LCD mutant mouse strain with a GOF develops a progressive neuromuscular phenotype, in the absence of LOF changes, thus showing that GOF is sufficient to initiate pathogenesis. Lastly, we show that splicing GOF and SEs occur also in another C-terminal ALS-causative *Tardbp* mutation and in fibroblasts derived from ALS patients carrying *TARDBP* mutations.

Our results shed light on novel aspects of TDP-43 biology and provide powerful new tools to gain insight into the early stages of TDP-43 neurodegeneration *in vivo*.

## Results

### Novel endogenous missense mutations in TDP-43 RRM2 and LCD

To study the effects of mutations in different domains of TDP-43 *in vivo*, at physiological levels, we selected two endogenous *Tardbp* mutations from a mouse allelic series found by screening mutagenised DNA archives from two large *N*-ethyl-*N*-nitrosourea (ENU) mouse mutagenesis programmes (Acevedo-Arozena *et al*, 2008; Gondo *et al*, 2010). The first mutation (F210I) is located in the important RNA recognition motif 2 (RRM2) of TDP-43; hence, we refer to this mutation as *RRM2mut* hereafter (Fig 1A).

The second mutation results in an M323K change in TDP-43 and falls within the C-terminal LCD where almost all known ALS-causing mutations are clustered—specifically within a 20 amino acid stretch that influences TDP-43 alpha-helix structure which is important for liquid phase separation, aggregation and protein self-interaction (Conicella *et al*, 2016). Importantly, we found in an *in vitro* turbidity assay that the M323K mutation containing C-terminal domain decreased the propensity for phase separation compared to wild type—similarly to the human ALS-causing mutation TDP-43 Q331K (Appendix Fig S1). Hereafter, we refer the M323K mutation strain as *LCDmut* (Fig 1A).

To generate cohorts of animals for *in vivo* analysis, we produced founder mice from the ENU programmes' sperm archives and generated litters by *in vitro* fertilisation; F1 mice were then backcrossed for at least five generations to eliminate non-linked ENU mutations and to place the mutations on congenic inbred backgrounds. We found both mutations, *RRM2mut* and *LCDmut*, were embryonic lethal on a congenic C57BL/6J background, supporting a significant impact of both amino acid changes on TDP-43 function (Appendix Table S1).

Genetic background effects can profoundly modify phenotype, and there are several reports in the literature of different backgrounds, or combinations of backgrounds, rescuing embryonic lethality (LeCouter *et al*, 1998). To determine whether we could produce homozygous animals by changing genetic background, we undertook a standard mouse genetics approach and backcrossed both *RRM2mut* and *LCDmut* lines onto C57BL/6J and onto DBA/2J for at least five generations and then worked with F1 C57BL/6J-DBA/2J intercrosses to produce homozygotes. We found the *RRM2mut* allele remained homozygous embryonic lethal and no live mice were detected. However, *LCDmut* homozygous mice were viable in this mixed genetic background (Appendix Table S1).

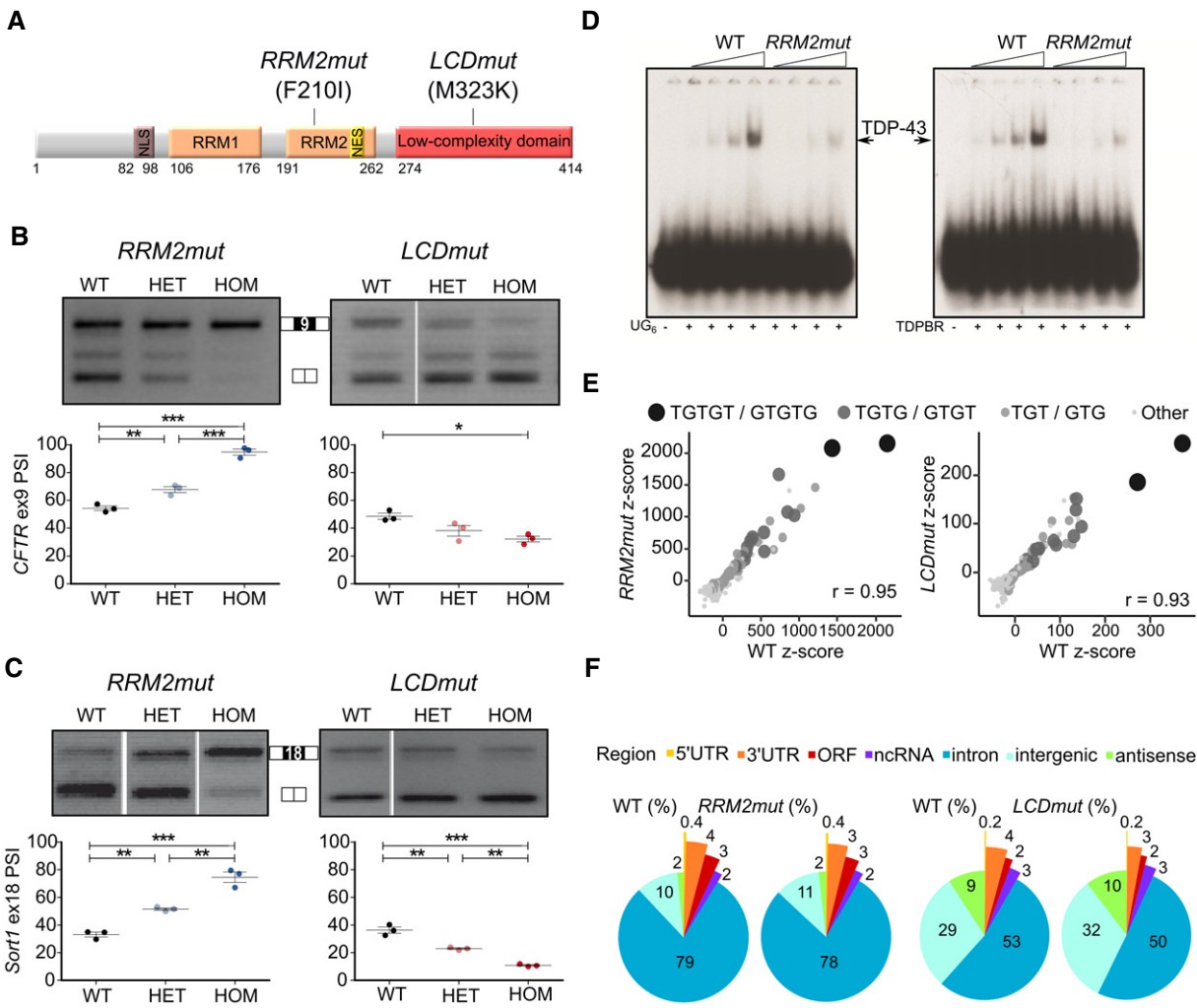

**Figure 1. Novel endogenous mutations *RRM2mut* and *LCDmut* have a loss- and gain-of-function effect on splicing.**

A   Diagram of TDP-43 illustrates the location of non-synonymous mutations used in this study.

B   Agarose gel and quantification of CFTR minigene splicing assay performed on MEFs from *RRM2mut* and *LCDmut* homozygous (HOM), heterozygous (HET) and littermate (WT) controls show an increase in exon inclusion in *RRM2mut* and a decrease in *LCDmut*. *RRM2mut* ANOVA $P < 0.0001$; *LCDmut* ANOVA $P = 0.0166$; Bonferroni multiple comparison tests are plotted as: \*$P < 0.05$; \*\*$P < 0.01$; \*\*\*$P < 0.001$; mean and SEM plotted; $N = 3$ per genotype.

C   Quantification of *Sortilin 1* RT–PCR products from *RRM2mut* and *LCDmut* MEFs shows exon 18 inclusion increases in *RRM2mut* and decreases in *LCDmut*. *RRM2mut* and *LCDmut* ANOVA $P < 0.0001$; Bonferroni multiple comparison tests are plotted as: \*\*$P < 0.01$; \*\*\*$P < 0.001$; mean and SEM plotted; $N = 3$ per genotype.

D   EMSA assay performed with increasing quantities of recombinant WT and *RRM2mut* TDP-43 with radiolabelled $(UG)_6$ RNA and TDPBR RNA shows reduced RNA binding of recombinant *RRM2mut*.

E   Scatter plot of pentamer enrichment in TDP-43 iCLIP shows a canonical TDP-43 pattern of enrichment in TG/GT-containing pentamers in *RRM2mut* (left) and *LCDmut* (right) along with littermate WT controls.

F   Mapping of TDP-43 binding sites in *RRM2mut* (left) and *LCDmut* (right) along with littermate WT controls iCLIP datasets shows a similar distribution genome-wide.

Source data are available online for this figure.

## RRM2mut and LCDmut cause splicing LOF and GOF, respectively

To investigate the effects of these TDP-43 mutations on splicing in an endogenous physiological context, we performed the CFTR minigene assay in mouse embryonic fibroblasts (MEFs). *RRM2mut* caused a highly significant shift towards exon inclusion, similar to the well-documented effect of TDP-43 knockdown (Appendix Fig S2), showing a dose-dependent LOF (Fig 1B). Surprisingly, the

*LCDmut* showed an opposite effect, leading to an increase in exon *exclusion*, suggesting GOF (Fig 1B).

We then investigated by RT–PCR whether a similar effect was present in a well-characterised endogenous TDP-43 splicing target, exon 18 of *Sortilin 1*. Exon 18 inclusion was increased in *RRM2mut*, as previously described in TDP-43 knockdown (Polymenidou *et al*, 2011), but had an opposite change in *LCDmut*, further supporting the splicing GOF (Fig 1C).

Since TDP-43 protein levels appeared to be unchanged in both lines in MEFs (Appendix Fig S3), we performed electromobility shift assays (EMSA) to evaluate the RNA binding capacity of WT, *RRM2mut* and *LCDmut* recombinant TDP-43 (Buratti & Baralle, 2001). RNA binding was reduced in *RRM2mut* TDP-43, whereas it was unchanged with the *LCDmut* mutation (Fig 1D and Appendix Fig S4).

To address whether the reduced RNA binding capacity in the RRM2mut was associated with changes in RNA binding *specificity,* we performed TDP-43 iCLIP (Tollervey *et al*, 2011) on CNS tissue. As *RRM2mut* homozygotes are not viable, we performed iCLIP in embryonic brains for that strain. However, as homozygous *LCDmut* are viable on a hybrid background, we were able to perform iCLIP on adult brain—the most relevant age for the effects of the TDP-43 mutation. The iCLIP data cannot be used to draw quantitative binding conclusions, but in each case, the binding motifs and transcriptome-wide distribution of *RRM2mut* and *LCDmut* TDP-43 binding sites within exons, introns, UTRs and intergenic regions showed no changes from wild-type TDP-43 (Fig 1E and F). Thus, *RRM2mut* is a hypomorphic allele, as it shows reduced RNA binding affinity whilst maintaining the typical TDP-43 sequence recognition and leads to a splicing loss of function. *LCDmut* instead induces a splicing gain of function whilst maintaining normal RNA binding capacity and specificity.

## RRM2mut and LCDmut show opposing effects transcriptome-wide and counteracting biological effects

We next sought to directly compare our two mutants to a *bona fide* TDP-43 LOF. We could not use TDP-43 null mice as they die before E6.5, so we performed RNA-seq on MEFs homozygous for the *RRM2mut* and the *LCDmut* alleles comparing them each to their wild-type controls, in parallel with wild-type MEFs in which we induced TDP-43 knockdown with shRNA (*Tardbp-shRNA*) comparing them to scramble sequence shRNA (Appendix Fig S5). Two-way comparisons showed that *RRM2mut* aligns with *Tardbp-shRNA,* with all significantly differentially expressed exons in both datasets up- or downregulated in agreement. Conversely and strikingly, *LCDmut* showed all of the significant differentially expressed exons changing in the opposing direction to *RRM2mut* and to *Tardbp-shRNA* (Fig 2A, Appendix Table S2).

These analyses identified splicing targets that are affected in opposing directions by *RRM2mut* and *LCDmut,* along with unique splicing events that occur within each mutant. To investigate these effects further, we crossed the two mutant lines, on a C57BL/6J background, and generated compound heterozygotes. Whilst *RRM2-mut* and *LCDmut* are both homozygous lethal in this background and produced 52 and 400 splicing changes, and 883 and 34 expression changes respectively, the compound heterozygous *RRM2/ LCDmut* mice were viable (although 8.8% of the mice instead of the 25% expected by Mendelian ratios, pointing to an incomplete rescue) and remarkably had only one significant splicing change and eight differentially expressed genes (FDR < 0.05)—indicating that *RRM2mut* LOF and *LCDmut* GOF complement each other and that the combination of mutations results in viable animals (Fig 2B and C).

Overall these results show that the two mutations have opposing LOF/GOF effects on splicing, providing a novel

mammalian system to specifically assess the role of TDP-43 LOF/ GOF *in vivo.*

## TDP-43 LCD mutations cause novel splicing events, the skipping of constitutive exons: skiptic exons

Given the relevance of LCD mutations to ALS, we focused our molecular analysis on adult spinal cord RNA-seq from 1-year-old *LCDmut* mice; we found 523 significantly differentially spliced events compared to WT littermate controls. Differential splicing events were not biased towards lowly expressed genes (Appendix Fig S6). When we analysed the distribution of splicing changes between the splicing event categories of cassette exons, retained introns, alternate 3′ and 5′ junctions, alternate first and last exons, and mutually exclusive exons, alterations in cassette exons were the most frequent events (46%; Fig EV1A), with a notably strong preference towards exon skipping events (in 79% of cassette exons; Fig EV1B). Motif enrichment analysis within 100 bp of each cassette exon:intron boundary identified the TDP-43 binding motif, UGUGUG, *only* for exon exclusion ($P = 5.2E-28$; Fig EV1C). Therefore, exon *skipping* is associated with TDP-43 direct RNA binding in *LCDmut* (Fig EV1G and H). This is in stark contrast to what we found in RNA-seq from our *RRM2mut* LOF line, where, consistent with previously published TDP-43 LOF datasets, the exon *inclusion* events have a stronger association with the UGUGUG TDP-43 binding motif (Fig EV1D–F).

We further analysed the skewed preference in *LCDmut* for exon skipping and identified a novel previously unreported effect: the skipping of otherwise constitutively expressed exons. We termed these novel events *skiptic exons* (SEs) and to define them we used the following per cent spliced in (PSI) criteria—exons that were included in WT (PSI > 0.95) and skipped in mutants (ΔPSI decrease of > 0.05).

We identified 47 SEs in 44 genes in our *LCDmut* RNA-seq data and validated 7 with RT–PCR. Mean and median PSI of these events in wild-type littermates were 0.98 and 0.99, mean and median ΔPSI in *LCDmut* were −0.12 and −0.09, and *LCDmut* exon exclusion mean and median fold-changes were 17.6 and 8.6, respectively (Fig 3A and C, Appendix Table S3). Crucially, SEs found in *LCDmut* are absent in *RRM2mut,* as exemplified by *Herc2* sashimi plots (Fig 3B), with the exception of one in *Tsn,* which is present also in compound heterozygous *RRM2mut/ LCDmut* samples. To exclude possible random sampling effects, we performed splicing analysis on 50 random sample permutations showing that the identification of SEs is not due to random effects (Appendix Table S4 and Appendix Fig S7).

Intriguingly, SEs represent the opposite phenomenon to recently described cryptic exons (CEs), in which TDP-43 LOF induces the inclusion of otherwise constitutively repressed exons (Ling *et al*, 2015; Humphrey *et al*, 2017). We defined cryptic exons (CEs) as exons that are significantly de-repressed using the reverse criteria from the above skiptic exons: repressed in WT (PSI < 0.05) and present in mutants (ΔPSI of > 0.05). We identified 33 CEs in the *RRM2mut* line and validated them by RT–PCR, but only three in the *LCDmut* (Figs 3A and EV2A and C). CEs identified in *RRM2mut,* with the exception of one event in *Reep3,* are absent in *LCDmut,* as exemplified by sashimi plots of *Adnp2* (Fig EV2B). Interestingly, *Reep3* CE is present also in our *RRM2mut/LCDmut* compound heterozygotes, suggesting a possible distinct mechanism for this event.

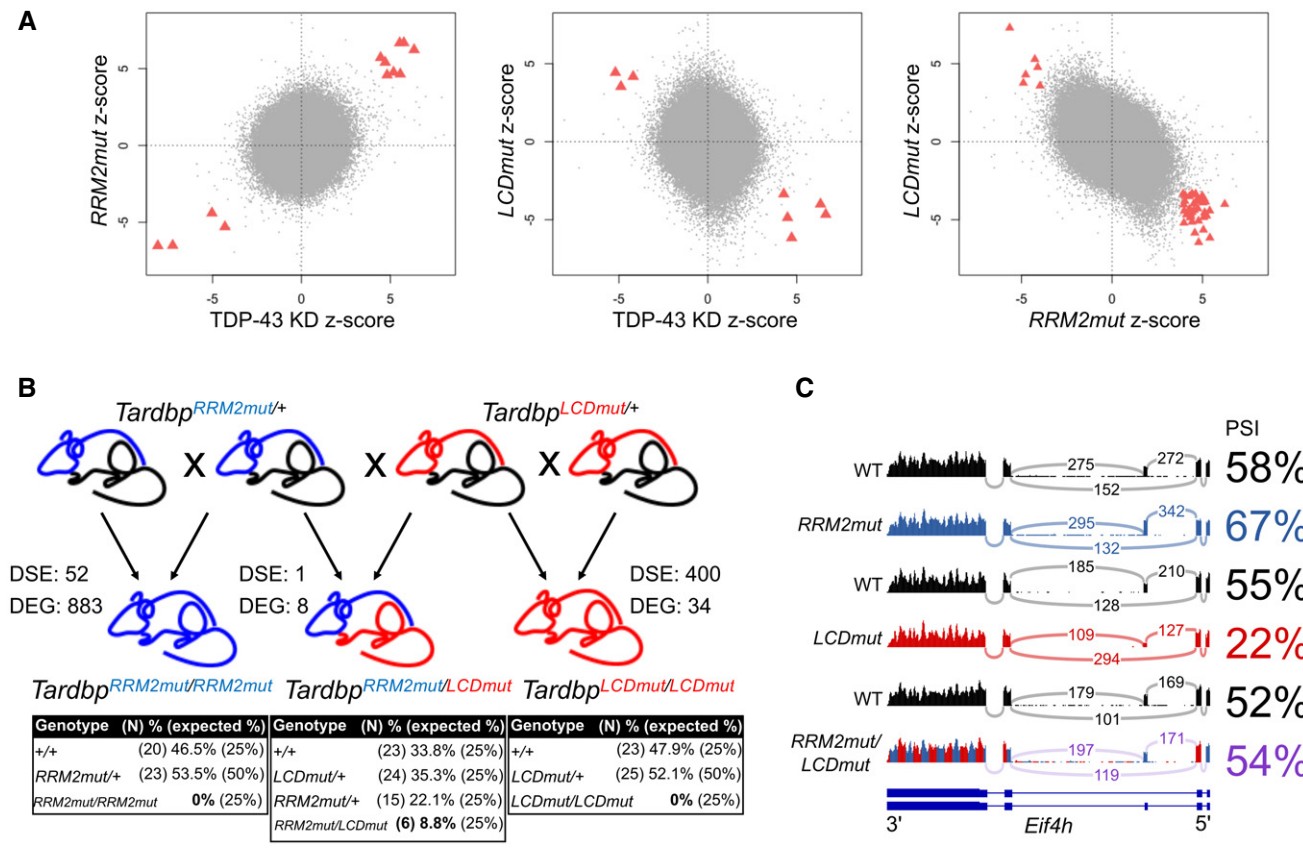

**Figure 2.   TDP-43 *RRM2mut* and *LCDmut* show opposing effects transcriptome-wide and counteracting biological effects.**

A   Two-way comparisons of MEF single exon differential expression changes between *RRM2mut*, *LCDmut* and TDP-43 knockdown (KD). *N* = 3 for each RNA-seq condition. Z-scores for single exons are plotted (significantly changed exons—FDR < 0.1, DEXSeq—in both datasets are highlighted in red and defined in Appendix Table S2).

B   Diagram illustrating the *RRM2mut* and *LCDmut* intercross, with differentially spliced exons (DSE, FDR < 0.01, DEXSeq; *N* = 3 for each RNA-seq genotype), differentially expressed genes (DEG, FDR < 0.05, DESeq2) and significant increase in survival of compound heterozygous (*P* < 0.05). N, numbers of animals produced per genotype; %, percentage of mice observed per genotype; expected %, percentage of mice expected per genotype.

C   Sashimi plots illustrate the unchanged *Eif4h* splicing in compound heterozygous mice compared to opposing altered splicing in *LCDmut* and *RRM2mut*. PSI determined by SGSeq are plotted.

Overall these results show that, whilst LOF and GOF act in opposing direction on a shared pool of alternatively spliced exons, GOF acts on a distinct set of genes inducing the skipping of constitutive exons (SEs) and LOF affects a separate set of genes where it induces the inclusion of CEs (Fig 4).

### Skiptic exons are linked to direct TDP-43 RNA binding

To determine whether TDP-43 RNA binding is associated with SEs, we used the combination of RNA-seq and iCLIP data. RNA maps show an enrichment of TDP-43 binding sites within SEs, and in their downstream introns, and this pattern parallels that of CEs in *RRM2mut* (Fig 5A and B). Appendix Table S5 illustrates how TDP-43 binding to SEs (66.0%) is enriched compared to other differentially alternatively spliced exons (18.8%), alternatively spliced exons (6.4%), constitutive exons (7.4%) and all GENCODE mouse exons (5%). The similarity of TDP-43 binding pattern distribution around SEs induced by GOF and CEs induced by LOF illustrates how these binding sites play a role in regulating exon inclusion/exclusion and

strongly support a direct role for TDP-43 RNA binding in these novel GOF events.

### SEs are highly conserved and lead to impaired transcript levels

To further characterise the SEs, we investigated their conservation, and interestingly, in contrast to CEs, conservation of SEs is high between species (Fig 5C). We also found that the novel skipping events produced by the *LCDmut* lead to frameshift or premature termination codons (PTC) in 15/48 transcripts, resulting in likely unstable transcripts. Therefore, we compared expression of transcripts containing SEs to transcripts carrying other differentially regulated cassette exons, and all other transcripts expressed at similar levels in our RNA-seq dataset; we found the transcripts that included SEs were significantly more downregulated (Fig 5D).

Furthermore, the proportion of predicted unstable transcripts significantly increases (*P* < 0.05) from 18% in SEs showing no expression change to 54% in significantly downregulated SEs

**Figure 3.  Skiptic exons—novel splicing events induced by TDP-43 GOF.**

A    Volcano plot of *LCDmut* RNA-seq shows significantly alternatively spliced exons that have either a PSI of < 0.05 and a ΔPSI of > 0.1 (CE, red) or a PSI of > 0.95 and a
ΔPSI of < −0.1 (SE, blue).
B    Sashimi plots show presence of a SE in *Herc2* in *LCDmut*, absent in *RRM2mut* and WT controls.
C    Diagram illustrating SEs (top), representative lanes for acrylamide capillary traces (middle), and quantification (bottom) of RT–PCR (*n* = 4) validations for seven SEs. *T*-test: *Ankrd42 P* = 9.286e-7; *Herc2 P* = 0.001269; *Pacrgl P* = 0.001487; *Pex16 P* = 3.29e-5; *Plod1 P* = 9.66e-5; *Slc6a6 P* = 2.077e-5; *Ube3c P* = 0.000263. Plotted *P*-value:
**\*\*P* < 0.01; \*\*\*P* < 0.001. Mean and SEM plotted.

Source data are available online for this figure.

(Fig 5E). These results show that SEs have an impact on expression levels *in vivo*.

## LCDmut induces a progressive neuromuscular phenotype and affects autoregulation of *Tardbp* mRNA

To investigate the physiological effect of our LCD mutation, we aged wild-type, heterozygous and homozygous littermates up to 24 months and found a progressive grip strength deficit in homozygous *LCDmut* male and female mice after 12 months of age (Fig 6A and Appendix Fig S8). We analysed *in vivo* muscle force and found a significant 38% reduction in homozygous *LCDmut* mouse tetanic

strength in tibialis anterior muscles at 24 months of age (Fig 6B). We performed a physiological analysis of motor unit survival (MUNE) to estimate motor units innervating the extensor digitorum (EDL) muscle at 24 months. We found a significant 15% reduction in homozygous *LCDmut* motor units compared to littermate controls (Fig 6C), showing that *LCDmut* mice develop a progressive motor dysfunction. Importantly, motor neuron (MN) counts performed on the sciatic MN pool at L4–L5 showed a 28% reduction in homozygous *LCDmut* when compared to littermate controls (*P* < 0.05; Fig 6D).

Immunohistochemistry investigations of the spinal cord at 18 months of age showed that homozygous *LCDmut* mice develop widespread p62- and ubiquitin-positive inclusions in the ventral

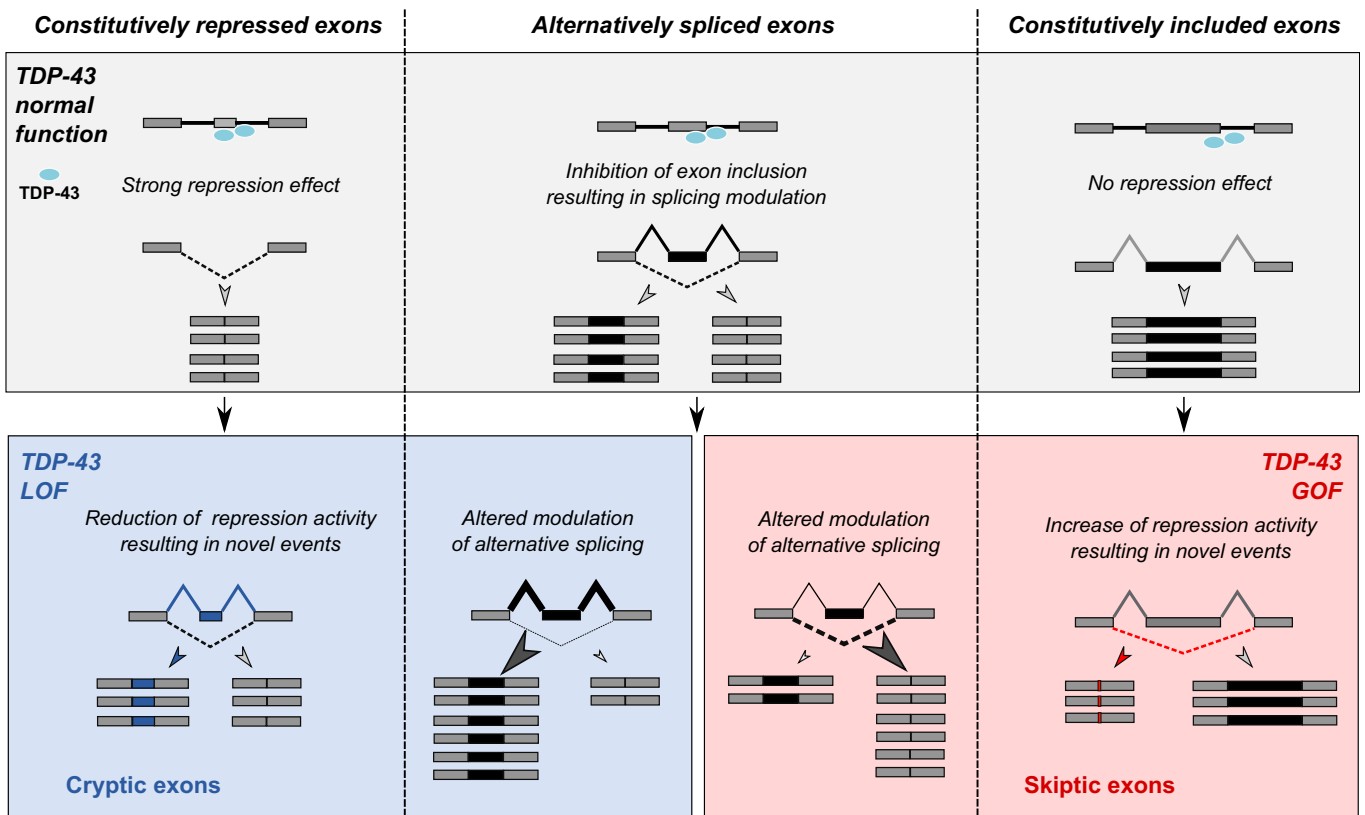

**Figure 4.   Diagram illustrating the effect of TDP-43 loss and gain of function on events where it acts to inhibit exon inclusion.**

TDP-43 has been shown to repress the inclusion of exons by binding to the pre-mRNA sequence of exons or to the downstream introns. In normal conditions, TDP-43 binding contributes to fully repress the inclusion of exons to mature transcripts (top left); it contributes to regulate the inclusion of alternatively spliced exons (top centre); and it also binds to exons and their downstream introns without an effect on exon exclusion (top right). TDP-43 LOF alters the balance of TDP-43 regulated alternative splicing and induces the de-repression of normally excluded exons, cryptic exons (bottom left). TDP-43 GOF affects TDP-43 regulated alternative splicing in the opposite manner and also induces the exclusions of otherwise normally constitutive exons, cryptic exons (bottom right).

regions (Fig 6E and Appendix Fig S8), a hallmark of neurodegeneration. We investigated whether this phenomenon was widespread in the CNS, and brain immunohistochemistry showed an increase in p62 pathology in *LCDmut* brainstem motor nuclei, whilst no change was detectable in other brain regions including the hippocampus (Appendix Fig S8). Intriguingly, although p62 inclusions are preferentially enriched in areas where MN is located (spinal cord ventral horns and brainstem motor nuclei), they do not appear to be localised to the cytoplasm of MN cell bodies.

We then investigated TDP-43 expression and whether TDP-43 was mislocalised and aggregated in the spinal cord. Surprisingly, we found *Tardbp* mRNA levels were significantly upregulated in a dose-dependent manner in heterozygous and homozygous *LCDmut* adult spinal cord (Fig EV3A) and this was consistent with changes in the well-described TDP-43 autoregulation mechanism, in which splicing of the *Tardbp* 3′UTR intron 7 (Appendix Fig S9) leads to a decrease in *Tardbp* levels. *Tardbp* intron 7 retention was significantly increased in a dose-dependent manner in heterozygous and homozygous *LCDmut* (Fig EV3B). Interestingly, the upregulation and autoregulation changes were also present at embryonic stage showing how these are induced by the presence of the mutation and are not a consequence of the neurodegenerative process. However, TDP-43 protein levels were not significantly changed (Fig EV3C)

and immunohistochemistry showed the absence of TDP-43 cytoplasmic inclusions and no nuclear depletion (Appendix Fig S8).

Of note, we also aged *RRM2mut* heterozygous animals up to 2 years and did not detect any of the above motor phenotypes or p62 pathology.

In summary, *LCDmut* gives rise to a progressive neuromuscular phenotype *in vivo*. The lack of nuclear depletion suggests that TDP-43 LOF is dispensable for initiation and development of this neuromuscular phenotype in *LCDmut* mice. Indeed, typical molecular LOF features, such as CEs and downregulation of long introns (Polymenidou *et al*, 2011), are absent in *LCDmut* mice (Figs EV2 and EV4).

Thus, *LCDMut* GOF, in the absence of TDP-43 aggregation, mislocalization and LOF, can directly lead to neurodegenerative phenotypes when expressed at physiological levels, suggesting TDP43 GOF plays a key role in driving TDP-43-related ALS.

### Splicing GOF and skiptic exons are found in a novel ALS-causative TDP-43 Q331K knock-in mouse model and in patient-derived cells

*LCDmut* is located in the hotspot for ALS-causing mutations and shares similarities *in vitro* with human pathogenic mutations; however, as the M323K mutation has not been described in humans

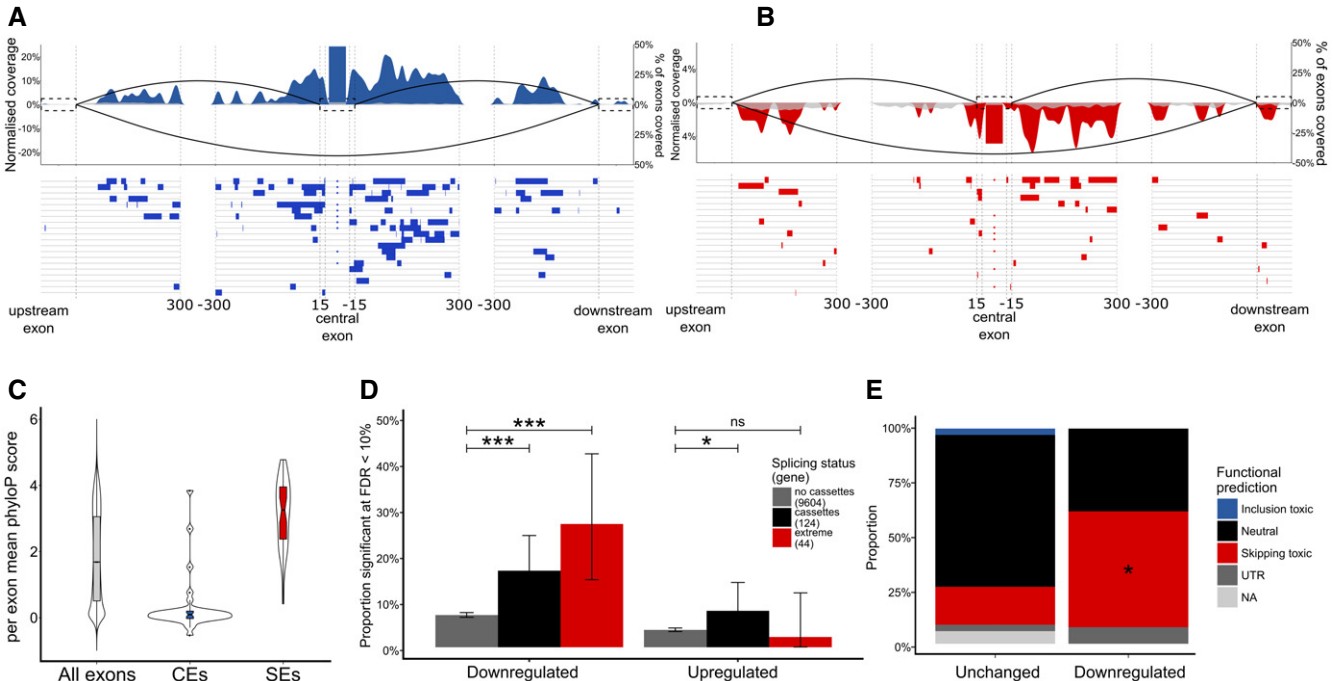

**Figure 5. SEs are induced by TDP-43 RNA binding and determine transcript dysregulation.**

A, B    RNA maps show binding distribution of TDP-43 iCLIP clusters (positive and negative values on *y*-axis for increased and decreased ΔPSI respectively) in CEs in *RRM2mut* (A) and SEs in *LCDmut* (B); red and blue indicate when cluster coverage is increased above background set of non-regulated cassette exons (grey). Below, binding sites for the top 20 most significant CEs (A) and SEs (B) are plotted.

C    Per-exon mean PhyloP conservation scores of RRM2mut CEs, LCDmut SEs and all annotated mouse exons. Scores are displayed as violin plots to show the full distribution of data points with overlaid boxplots to show the median and quartiles. Notches represent the 95% confidence interval of the median, and whiskers represent the minimum and maximum values that fall within 1.5 times the interquartile range. Outliers are plotted as black dots.

D    The relationship between splicing status and differential expression in LCDmut. Significantly downregulated genes (FDR < 0.1) are enriched in SEs compared to genes expressed at a similar level (*P* = 1.19e-6), as well as non-skiptic cassette exons (*P* = 6.16e-5). Upregulated genes are mildly enriched in non-skiptic cassette exons (*P* = 0.029) but not in SEs (*P* = 0.88). All *P*-values generated from a binomial test.

E    Damaging and unstable changes are more frequent in downregulated SEs (53.8%) versus other SEs (17.6%). **P* = 0.034; chi-squared test.

with ALS/FTD, we asked whether the molecular phenomena we identified are induced by other LCD disease-causing mutations. Thus, we generated a novel knock-in (KI) model via CRISPR/Cas9 genome editing carrying a Q331K mutation (Fig 7A) and we assessed splicing effects by performing RT–PCR on primary cells and adult spinal cord from these mice. As identified in *LCDmut,* we confirmed a gain of splicing function in Q331K mice leading to increased exon skipping in the *CFTR* minigene and in endogenous *Sortilin 1* and *Eif4h* and increased exon inclusion in *Kcnip2* (Fig 7B and C). Furthermore, we were able to confirm that Q331K also induces the presence of SEs in *Ube3c*, *Pex16* and *Pacrgl* (Fig 7D).

An important difference between the SEs and CEs is the significantly higher conservation of SEs across species (Fig 5C), leading to the question of whether these events are present in patients carrying ALS-causing *TARDBP* mutations. We used fibroblasts from four different ALS patients carrying the *TARDBP* G298S and A382T pathogenic variants (Appendix Table S6) to investigate whether SEs occur in patient cells. We tested the seven SEs we had validated in our mice and found two to be present and significantly changed in patient-derived mutant fibroblasts when compared to controls (Fig 7E and Appendix Fig S10). Overall, these findings in a novel knock-in mouse line and in patient-derived cells support the finding that splicing GOF is induced by ALS-causing mutations, although

further studies will be necessary to assess how widespread the GOF induced by ALS-causing mutations is.

# Discussion

Both TDP-43 gain and loss of function have been proposed as possible crucial mechanisms in the pathogenesis of ALS, a devastating incurable neurodegenerative disorder, and both reduced TDP-43 expression and TDP-43 overexpression have detrimental effects in mice (Arnold *et al*, 2013; Yang *et al*, 2014).

Here, we present novel mouse lines with point mutations within endogenous *Tardbp*. As TDP-43 is very tightly autoregulated, and in transgenic mice, even small changes in WT TDP-43 levels induce vast LOF/GOF transcriptome alterations, our mice provide the unique possibility of investigating the effect of mutations on TDP-43 function in absence of confounding effects due to overexpression.

We show with a well-established splicing assay and RNA-seq that the *RRM2* mutation leads to a dose-dependent loss of TDP-43 function, whereas the *LCDmut* leads to a gain of function.

The LOF effect of *RRM2mut* was expected, because the mutation reduces TDP-43 RNA binding capacity; thus, this mouse model

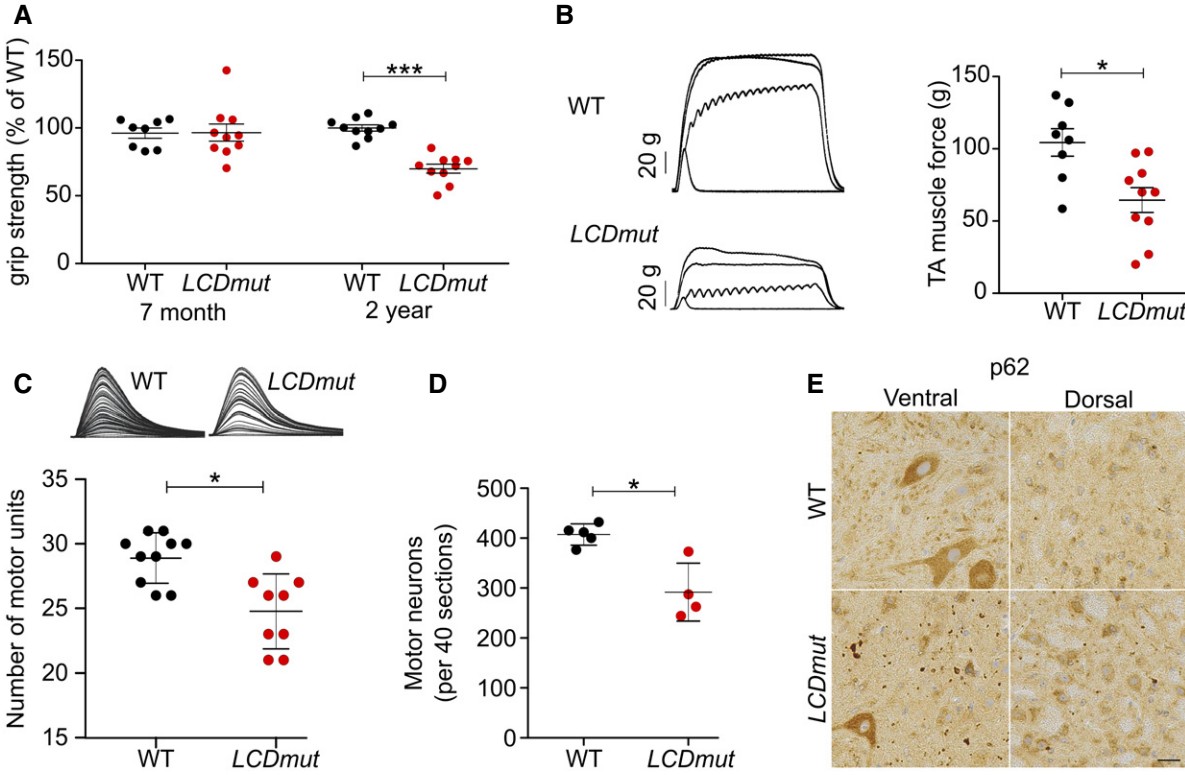

**Figure 6. *LCDmut* develops a neuromuscular and neurodegenerative phenotype.**

A *LCDmut* homozygous female mice show significant 30% grip strength deficit at 2 years, whilst no change is present at 7 months. $P = 0.829$ (7 months) and ***$P < 0.0001$ (2 years), Mann–Whitney test. Mean and SEM plotted; $N = 8–10$, as plotted.

B TA muscle strength is decreased in 2-year-old *LCDmut* mice; representative traces are shown, *x*-axis, time; *y*-axis, force (*top*) and tetanic force from 5 *LCDmut* (64.5 g ± 26.8 SD) and WT (104.4 g ± 26.1 SD) female mice are plotted (*bottom*). *$P = 0.0113$, Mann–Whitney test $N = 8–10$, as plotted.

C Surviving motor units for the EDL muscle are reduced in 2-year-old *LCDmut* (24.7 ± 2.8 SD) compared with WT (28.9 ± 1.9 SD); representative traces where each twitch trace recording is a single motor unit (*left*). *$P = 0.0057$, Mann–Whitney test. $N = 9–10$, as plotted.

D Counts of MNs from L4–L5 show a significant reduction (28% reduction; *$P = 0.0159$, Mann–Whitney test, mean and SD plotted) $N = 4–5$, as plotted.

E *LCDmut* mice show p62 pathology in the spinal cord ventral horns at > 18 months. Scale bar indicates 10 μm.

Data information: Two-tailed Mann–Whitney *t*-test *P*-values are plotted as: *$P < 0.05$; ***$P < 0.001$.

---

provides a novel tool to study chronic LOF *in vivo* and notably, in contrast to homozygous TDP-43 null mice, which do not survive beyond embryonic day 6 (E6.5; Kraemer *et al*, 2010; Sephton *et al*, 2010; Wu *et al*, 2010; Ricketts *et al*, 2014), *RRM2mut* embryos survived to at least E18.5, allowing us to study embryonic tissue and primary cell cultures.

Conversely, the splicing GOF effect produced by *LCDmut* is unexpected and, although GOF effects had been previously described in transgenic models where TDP-43 is overexpressed, the *physiological setting* of the mutations in these mice shows for the first time that C-terminal TDP-43 mutations result in a splicing GOF.

Compound heterozygous animals from intercrosses between the *RRM2* and *LCDmut* strains show a partially rescued mouse embryonic lethality and a rescue of the majority of splicing changes demonstrating that the LOF and GOF compensate each other *in vivo*. The partial rescue of the lethality phenotype may reflect the non-symmetry of the LOF/GOF changes.

Studying primary cells and tissue from our two models side by side has allowed us to dissect LOF and GOF and to identify common targets and specific features of the two processes. Genome-wide

analysis identified splicing targets that are affected in opposing directions by *RRM2mut* and *LCDmut*. Interestingly, also unique splicing events occur within each mutant.

We have identified that GOF induces the skipping of normally constitutively expressed exons, here named skiptic exons. These events are induced by the TDP-43-dependent splicing GOF mechanism we describe, but occur within a set of genes previously unknown to be regulated by TDP-43. Intriguingly, these events appear to mirror a previously described phenomenon, whereby TDP-43 LOF induces the inclusion of cryptic exons. Importantly, combining our RNA-seq and iCLIP data, we show that the TDP-43 binding pattern surrounding CEs and SEs is very similar, supporting a role for TDP-43 binding in regulating both events. TDP-43 is involved in the exclusion of these exons; therefore, GOF determines excessive exclusion, whilst LOF de-represses exons that are normally not included (Fig 4). Although TDP-43 RNA binding likely mediates the splicing GOF, it is unclear what determines the GOF and whether changes in protein–protein interactions caused by the C-terminal mutations could contribute to the GOF. This is a possibility that will require further study. Whilst TDP-43 binds to thousands

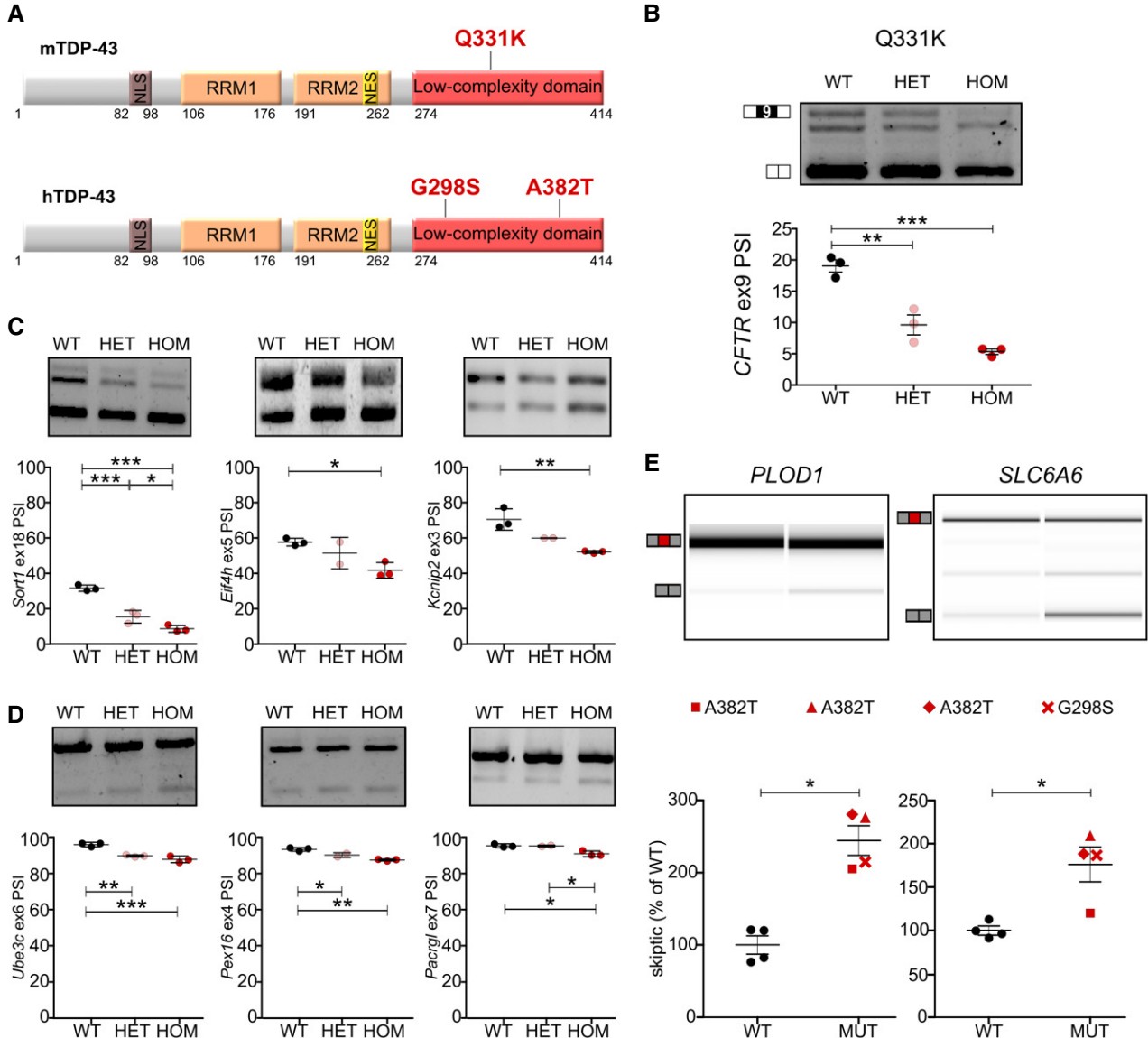

**Figure 7.** TDP-43 splicing GOF and skiptic exons are present in a novel knock-in mouse model and in patient-derived cells.

A Diagram illustrating location of ALS-causative mutations in *Tardbp* (Q331K, mouse knock-in) and *TARDBP* (G298S and A382T, patient-derived fibroblasts).

B Agarose gel and quantification of CFTR minigene splicing assay performed on MEFs from *Tardbp Q331K* homozygous (HOM, *N* = 3), heterozygous (HET, *N* = 3) and littermate (WT, *N* = 3) controls show a decrease in exon inclusion in mutant vs. WT controls (*P* = 0.0003). Middle band represents the activation of a cryptic site in the minigene.

C PSI quantification from agarose gels shows significant *Sortilin 1* (*P* < 0.0001) splicing changes in MEFs from Q331K mice (similarly to *LCDmut* in Fig 1B) and significant *Eif4h* (*P* = 0.0290) and *Kcnip2* (*P* = 0.0053) GOF changes in 3-month Q331K spinal cord (WT *N* = 3, Het *N* = 2–3, HOM *N* = 3).

D Significant increase and quantification of *Ube3c* (*P* = 0.0004), *Pex16* (*P* = 0.0010) and *Pacrgl* (*P* = 0.0086) SEs in Q331K mouse spinal cord (WT *N* = 3, Het *N* = 2–3, HOM *N* = 3).

E *PLOD1* and *SLC6A6* SEs are present and significantly increased in fibroblasts derived ALS patients carrying TARDBP pathogenic mutations. *PLOD1* (*t*-test; *P* = 0.01713); *SLC6A6* (*t*-test; *P* = 0.02224); *N* = 4.

Data information: Representative gels (top) include *TARDBP* G298S for *SLC6A6* and *TARDBP* A382T for *PLOD1*; quantification (bottom)—ratio between SE and the exon inclusion bands, after normalizing the mean of healthy controls to 100%, is used instead of PSI, as amplification was performed using a three primer PCR. For (B, C and D), *P*-value from ANOVA is indicated in legend and Bonferroni multiple comparison test *P*-values are plotted as: *\**P* < 0.05; *\*\**P* < 0.01; *\*\*\**P* < 0.001. Mean and SEM are plotted.

Source data are available online for this figure.

of constitutive exons throughout the transcriptome, only 44 undergo skipping in the presence of *LCDmut* induced GOF; this is likely due to the cooperative nature of splicing and that TDP-43 is only one of many factors involved in the regulation of the inclusion/exclusion of the thousands of exons it binds to (Mohagheghi *et al*, 2016).

TDP-43 acts to regulate alternative splicing, but critically, here we uncover how GOF and LOF have an effect on exons that are usually not alternatively regulated, but are instead constitutively included or excluded. Thus, CEs and SEs are not a novel splicing mechanism, but the effect of LOF and GOF, respectively, on a subset of constitutively included and excluded exons. This phenomenon is important as it highlights how TDP-43 GOF and LOF determine alterations in distinct subsets of genes (Fig 4).

We find downregulation of the overall expression of genes harbouring skiptic exons in *LCDmut* mice, showing that these novel events lead to changes in expression levels. Interestingly, 7/44 identified SEs are within genes involved in the ubiquitin proteasome pathway, including five E3 Ubiquitin ligases: *Herc2*, *Ube3c*, *Wwp1*, *Ttc3* and *Wsb1*. Overall, these data raise the question of whether these events have the potential to alter CNS proteostasis, a central mechanism involved in neurodegeneration (Ling *et al*, 2013).

*In vivo*, *LCDmut* mice develop progressive motor deficits, muscle weakness, reduced motor unit numbers and MN loss accompanied by p62- and ubiquitin-positive ventral horn inclusions. By 24 months, the mice do not develop paralysis or show premature death, which are typical features of ALS; instead, the progressive phenotype they develop resembles the initial stages of disease. This is consistent with other ALS models, as overexpression of ALS proteins induces rapid severe phenotypes in transgenic mice (Gurney *et al*, 1994; Mitchell *et al*, 2013), whilst mutations in endogenous *Sod1* and *Fus* show slower milder neuromuscular phenotypes (Joyce *et al*, 2014; Devoy *et al*, 2017; Scekic-Zahirovic *et al*, 2017).

The progressive neuromuscular phenotype accompanied by MN loss supports the relevance to disease of the *LCDmut* mice, that carry a mutation which shares similarities *in vitro* with the ALS-causing Q331K mutation, but has not yet been identified in ALS patients. Here, we use the Q331K knock-in strain, which is currently being aged and fully characterised, to show that splicing GOF is also present in this ALS-causing C-terminal TDP-43 mutant. The effect observed in Q331K was significant, but smaller than in *LCDmut*, possibly due to the younger age of the analysed mice or the different genetic background. Lastly, we were able to move from a classic investigation of mechanism in CNS of mouse models, to examining the same phenomena in patient-derived cells. We validated two of the skiptic exons in fibroblasts derived from patients carrying *TARDBP* ALS-causative mutations—these results warrant more comprehensive transcriptomic studies in patient-derived material, which are currently lacking for *TARDBP* ALS mutations.

Importantly, the motor phenotype in homozygous *LCDmut* mice develops without TDP-43 nuclear depletion, TDP-43 inclusions or features of TDP-43 LOF, such as the inclusion of cryptic exons or downregulation of long intron genes, showing that these phenomena are not necessary for the initial stages of neurodegeneration, as also demonstrated by others (Arnold *et al*, 2013).

An abnormal upregulation of *TARDBP* has been shown in human TDP-43-ALS post-mortem end-stage brain, likely due to the absence of nuclear TDP-43 protein, and a vicious cycle of increased expression has been proposed to play a role in disease (Koyama *et al*, 2016). Here, unexpectedly, we found an imbalance in the autoregulation and the upregulation of *Tardbp* levels in *LCDmut*, throughout life, although protein levels were not significantly increased. It is important to note that in *Tardbp* heterozygous null mice, changes in TDP-43-dependent splicing occur in the absence of detectable

TDP-43 protein changes (Ricketts *et al*, 2014), and therefore, whether in *LCDmut* mice a subtle chronic increase in TDP-43 levels, below the detection threshold from Western blotting, could contribute to the observed GOF, will need to be investigated.

Intriguingly, the fact that both gain and loss of function induce *Tardbp* upregulation suggests the autoregulation imbalance may be a common process throughout disease and potentially represents a target for disease-modifying interventions. Importantly, our data show that TDP-43 GOF and LOF do not simply act on the same set of genes; rather, both are characterised by specific effects on gene regulation, producing a unique transcriptomic signature that may play distinct roles at different stages of disease pathogenesis. Indeed, whilst LOF likely plays a role at disease end stage, here we show that GOF is sufficient to initiate the neurodegenerative process.

In summary by working with mouse models with endogenous mutations in key TDP-43 domains and with patient-derived human fibroblasts, we separate GOF and LOF mechanisms and their effects, we provide a novel mouse model of ALS and identify an entirely unsuspected GOF acting on splicing of conserved constitutive exons. Finally, we show that, at least in this model, LOF is not necessary for initiating the neurodegenerative processes, whilst highlighting that the occurrence of splicing GOF is associated with the early stages of disease.

# Materials and Methods

### Mouse lines

The *Tardbp RRM2mut* and *LCDmut* lines were derived from the RIKEN BioResource Center and are available as BRC no. GD000108 and GD000110 from RIKEN BioResource Center, respectively.

The *Tardbp* Q331K line was generated in this study and is available upon request.

### Screening, identification and generation of ENU mouse lines

The RIKEN and MRC Harwell ENU-DNA archives were screened for coding mutations within *Tardbp*. The mutant lines were screened and established from both ENU mutant mouse libraries as previously described (Coghill *et al*, 2002; Sakuraba *et al*, 2005). Briefly, we screened the PCR-amplified fragments from a total of < 10,000 G1 individuals DNA samples by heteroduplex detection system. The G1 genomic DNA samples that exhibited the heteroduplex signals were subjected to Sanger sequencing to identify the ENU-induced mutations. We found seven *Tardbp* missense mutations: T4C, A42C, N45S, K160R, F210I, N312S and M323K. We selected F210I (*RRM2-mut*) and M323K (*LCDmut*) for further analysis, which were initially on a hybrid DBA/2JxC57BL/6J background. Mice were rederived via *in vitro* fertilisation (IVF) using frozen F1 sperm at RIKEN and maintained on C57BL/6J and DBA/2J backgrounds at MRC Harwell.

### Mouse housing, maintenance, genotyping and grip strength analysis

All animals were housed and maintained in the Mary Lyon Centre, MRC Harwell, under specific opportunistic pathogen-free (SOPF) conditions, in individually ventilated cages adhering to

environmental conditions as outlined in the Home Office Code of Practice. All animal studies were licensed by the Home Office under the Animals (Scientific Procedures) Act 1986 Amendment Regulations 2012 (SI 4 2012/3039), UK, and additionally approved by the Institutional Ethical Review Committee. Mice were euthanised by Home Office Schedule 1 methods.

Initial genotyping for the Q331K mutation was performed as follows: genomic DNA from F0 and F1 animals was extracted from ear clip biopsies using DNA Extract All Reagents Kit (Applied Biosystems). The targeted region was PCR-amplified using high-fidelity Expand Long Range dNTPack (Roche) and genotyping primers Geno_Tardbp_F1/R1. PCR products were further purified using gel extraction kit (Qiagen) and analysed by Sanger sequencing. PCR product amplified from DNA obtained from F0s that showed the correct point mutation (C>A inducing Q331K) was subcloned using Zero-Blunt PCR cloning Kit (Invitrogen), and 12 clones were analysed by Sanger sequencing.

A grip strength meter (BioSeb) was used to measure muscle strength. Total limb (fore and hind) were measured twice with an average value recorded, as previously (Ricketts *et al*, 2014).

## Mouse embryonic fibroblasts (MEFs) and silencing of TDP-43

Mouse embryonic fibroblasts were derived following established protocols from E14.5 embryos derived from intercrosses to obtain all genotypes within one generation, allowing to use littermates as controls. Primary cells were not tested for mycoplasma. In order to obtain TDP-43 knockdown in MEFs, we used shRNA from the GIPZ Lentiviral shRNA library (GE Dharmacon) with the following sequence shared between mouse *Tardbp* and human *TARDBP* (TGGATGAGACAGATGCTTC) in parallel with scramble control. In order to improve the percentage of MEFs expressing shRNA, 3 days after transfection, MEFs underwent 3-day treatment with puromycin at 4 μg/ml and 3 days at 2 μg/ml before being harvested in RIPA buffer for protein analysis and Trizol for RNA extraction. *Tardbp* levels were validated using *Tardbp* FAM TaqMan probes along with *Gapdh* and *Actb* VIC probes in separate reactions.

## CFTR minigene assay

Electroporation was used to transfect the CFTR minigene construct (Buratti *et al*, 2007) into MEFs of the different *Tardbp* genotypes. Briefly, an Amaxa system (Lonza) was used following manufacturer's instructions. After 24 h, MEFs were harvested and cell pellets snap-frozen. For this and any other validation of any splicing event, RNA was extracted following standard procedures via Trizol (Life Technologies) or columns (Qiagen), and cDNA produced via RT–PCR following manufacturer's instructions (Applied Biosystems, QuantaBio). CFTR exon inclusion was assessed via semiquantitative RT–PCR using primers specific against the minigene construct.

## Electrophoretic mobility shift assay (EMSA)

Mouse TDP-43 WT and the different mutations were expressed in *Escherichia coli* with a GST tag that was used to purify the different recombinant proteins. Different concentrations of recombinant proteins (0.25, 0.5, 1, 2 μg) or $UG_6$ repeats were used for different assays. The EMSA gel (5% acrylamide) was loaded with the recombinant protein in binding buffer together with $\gamma^{32}P$ labelled RNA ($UG_6$ repeats or *Tardbp* CLIP sequence). After the run was completed, the gel was dried (583 Gel Dryer Bio-Rad) and visualised using X-OMAT film or autoradiographic XAR film (F5763 Sigma/Kodak).

## RNA isolation, quality control, RT–PCR and RT–qPCR

For validation of splicing events from different sources, RNA was extracted from MEFs, embryonic E18.5 head or adult spinal cord. RNA quality was determined by RNA Integrity Number (RIN) measured using RNA ScreenTape and reagents (Agilent) on the 2200 Tapestation System (Agilent). RT–PCR protocol was conducted following manufacturer's instructions (Thermo, QuantaBio, Applied Biosystems). PCR for splicing events was conducted using 2X PCR Master Mix (Thermo) with specific primers spanning the differentially expressed exon (*RRM2mut* cryptic $n = 4$, *LCDmut* skiptic $n = 4$, Q331K splicing/skiptic $n = 3$). Products were electrophoresed on agarose gels with ethidium bromide (Sigma) or using D1000 Screen-Tape and reagents (Agilent) on the 2200 Tapestation system (Agilent). Results were analysed using PSI, calculated dividing the intensity of the exon inclusion band by the sum of both exon inclusion and exon exclusion bands. When analysing SEs in human fibroblasts, as a combination of three primers was used, we did not calculate PSI, but calculated the ratio between SE band and the exon inclusion band and expressed results after normalizing the mean of healthy controls to 100%. Primer sequences are available upon request.

## RNA-sequencing

RNA-seq was performed using polyA libraries as previously described (Wang *et al*, 2012; Joyce *et al*, 2016) and all RNA-sequencing generated for this study, along with read-length and average uniquely mapped reads are summarised in the table below. All mutants were compared to wild-type littermates. *N* refers to the number per genotype (control, heterozygous and homozygous).

All sequencing data are deposited in the NCBI Sequence Read Archive (reference: SRP133158).

## iCLIP

iCLIP was performed as detailed by König *et al* (2010). Dissected forebrain (*RRM2mut*) or adult brain (*LCDmut*) were homogenised in phosphate-buffered saline (PBS) and UV-crosslinked 4× at 100 mJ/cm$^2$ at a wavelength of 254 nm (*RRM2mut* $n = 2$, *LCDmut* $n = 1$–2). Protein concentrations were normalised using the DC Protein Assay (Bio-Rad), and the lysate was then sonicated for 30 s ×10 times. Lysates were then treated with 0.1 U/μl RNase I (Thermo) and Turbo DNase (Thermo). Anti-TDP-43 antibody (Sigma) was conjugated to Dynabeads Protein A (Thermo) and used to immunoprecipitate TDP-43 from the tissue lysate. The 3′ ends of attached transcripts were then dephosphorylated using T4 PNK (NEB) and the pre-adenylated L3-App adaptor ligated using RNA ligase (NEB). The 5′ end of transcripts were labelled with radioisotope ATP[γ−$^{32}$P] using T4 PNK. The labelled complexes were then resolved on NuPAGE 4–12% Bis-Tris gels (Invitrogen), transferred to nitrocellulose membrane and exposed to Fuji film. The protein–RNA complexes corresponding to TDP-43 size of 43 kDa and above were isolated, and the protein was digested with proteinase

K (Sigma). RNA fragments were reverse-transcribed using Super-Script III (Invitrogen) and size-resolved on Novex 6% TBE-urea gels (Invitrogen). Bands of size 75–95 nt and 95–200 nt were excised, further purified, circularised using Circligase (Epicentre) and digested with BamHI. cDNA fragments were amplified by PCR and further purified with Ampure XP beads (Beckman Coulter). The iCLIP libraries were quantified using the D1000 ScreenTape and reagents (Agilent) on the 2200 Tapestation System (Agilent) and sequenced using the standard Illumina protocol for 50-bp single-end sequencing (Huppertz *et al*, 2014).

### Immunohistochemistry

For analysis of spinal cord pathology, 1- and 2-year-old WT, *LCDmut* HET and HOM mice were perfused with saline followed by 4% PFA. The spinal cord was stored in formalin and then embedded in paraffin wax. 4-µm transverse sections were taken from the lumbar regions of the spinal cord. Staining for p62 (1:250, Abcam), TDP-43 (1:500, Novus Biologicals) and ubiquitin (1:5,000, Santa Cruz) was conducted using a Ventana immunohistochemical staining machine (Ventana). Following staining with the appropriate secondary antibodies, the sections were developed with 3,3-diaminobenzidine. All experiments were carried out using a positive [human frontotemporal dementia (FTD) cortex] and negative control.

### Immunoblot

Spinal cords from 2-year-old LCDmut HOM mice were homogenised in RIPA buffer with protease inhibitors. The homogenate was resolved using Western blot using NuPAGE 4–12% Bis-Tris gels (Invitrogen) using anti-TDP-43 (Proteintech). Quantification of the results was conducted on the Odyssey infrared imaging system (Li-Cor).

### Turbidity assay

Briefly, turbidity measurements were obtained on WT and mutant TDP-43 C-terminal fragments (267–414) thawed from frozen stock, desalted into 20 mM MES (pH 6.1) and diluted to 20 µM in a 100 µl final volume in 20 mM MES supplemented with 0–500 mM NaCl (Conicella *et al*, 2016). Turbidity at room temperature was quantified by measuring the optical density at 600 nm wavelength light at 5-min intervals over a 12-h time period. All measurements were recorded in triplicate in sealed 96-well clear bottom plates.

### Motor neuron survival

Mice were terminally anaesthetised with 4.5% chlorohydrate (1 ml/100 g of body weight), transcardially perfused with 4% PFA and the lumbar region of the spinal cord removed, post-fixed in 4% PFA and cryopreserved in 30% sucrose. Serial 20-µm transverse sections were cut on a cryostat, stained with gallocyanin, a Nissl stain, and the number of positively stained motor neurons in the sciatic motor pool of every third section between the L4 and L5 levels of the spinal cord was counted, as previously described (Joyce *et al*, 2016). Only large, polygonal neurons with a clearly identifiable nucleus and nucleolus were included in counts. This protocol avoids the possibility of counting the same cells twice in consecutive sections. A total of 40

sections were examined per spinal cord, and at least five female mice were analysed from each experimental group.

### Physiology analysis

Muscle force and motor unit number female WT and *LCDmut* littermates ($N$ = 5) were examined at 24 months of age, as described previously (Joyce *et al*, 2016). In summary, in anesthetised mice (4.5% chlorohydrate, 1 ml/100 g of body weight), the distal tendons of the tibialis anterior (TA) and extensor digitorum longus (EDL) muscles in both hindlimbs attached to isometric force transducers (Dynamometer UFI Devices, Welwyn Garden City, UK), and the sciatic nerve exposed and sectioned. Squarewave pulses of 0.02 ms duration at supra-maximal intensity were used to stimulate the nerve and elicit TA or EDL isometric contractions. Trains of stimuli at frequencies of 40, 80 and 100 Hz were used and were measured using Scope software. Motor unit numbers innervating the EDL was determined by using stimuli of increasing intensity, resulting in incremental increases in twitch tension due to successive recruitment of motor axons. The number of increments was counted giving an estimate of the number of functional motor units present in EDL muscle.

### Generation of *Tardbp* Q331K knock-in mice

The founder mutant Q331K mice were produced and characterised as per Mianné *et al* (2017). Specifically, the mutants were generated through C57BL/6J zygote pronuclear injection of CRISPR reagents, including the Cas9 mRNA, a sgRNA and a PAGE-purified ssODN donor template at 100, 50 and 50 ng/µl, respectively. Pups were genotyped by genomic DNA PCR amplification and Sanger sequencing of the targeted sequence. The mutant harbouring the intended nucleotide modification (C>A inducing Q331K) was then mated to a wild-type (WT) counterpart in order to assess germline transmission of the Q331K mutated allele. sgRNA was selected using online design tools in order to cut as close as possible to the targeted nucleotide change, having as few potential off-targets and having the minimal total number of potential off-targets (especially on the targeted chromosome 4). The ssODN donor oligo was ordered as PAGE-purified UltramerTM DNA oligo (IDT). The targeted nucleotide change disrupting the recognition site of the sgRNA selected, no additional modification was added to the ssODN donor oligo. Single-guide RNAs were synthesised from a plasmid containing the T7 promoter using MEGAshortscript kit (Ambion) following manufacturer's instructions and purified using MEGAclear kit (Ambion). RNA quality was assessed using a NanoDrop (Thermo Scientific) and by electrophoresis on 2% agarose gels containing ethidium bromide (Fisher Scientific). Cas9 mRNA was synthesised in-house from a plasmid containing mammalian codon optimised Cas9 sequence under T7 promoter. The plasmid was linearised and phenol–chloroform purified; mRNA was synthesised using Message Max T7 Arca Capped Message Transcription Kit (Cellscript), polyadenylated using poly(A) polymerase Tailing Kit (Epicentre) and purified using MEGAclear kit (Ambion). mRNA quality was assessed using a NanoDrop (Thermo Scientific) and by electrophoresis on 2% agarose gel containing ethidium bromide (Fisher Scientific).

Pronuclear microinjection was performed as per Gardiner & Teboul (2009), employing a FemtojJet (Eppendorf) and C57BL/6J embryos. Specifically, injection pressure (Pi) was set between 100

and 700 hPa, depending on needle opening; injection time (Ti) was set at 0.5 s and the compensation pressure (PC) was set at 10 hPa. Microinjection buffer (MIB; 10 mM Tris–HCl, 0.1 mM EDTA, 100 mM NaCl, pH 7.5) was prepared and filtered through 2-nm filter and autoclaved. Cas9 mRNA, sgRNA and ssODN were diluted and mixed in MIB to the working concentrations of 100, 50 and 50 ng/μl, respectively. Injected embryos were re-implanted in CD1 pseudo-pregnant females. Host females were allowed to litter and rear F0s.

All sequences used to generate and characterise the mutants are the following: sgRNA: GGCAGCGTTGCAGAGCAGTT GGG; ssODN donor (targeted nucleotide change in bold and underlined): CCCGACTGGTTCTGCTGGCTGGCTAACATGCCCATCATACCCCAAC TGCTCTTCAACGCTGCCTGAGCCGCAGCCATCATCGCTGGGTTAAT GCTAAAAGCACCAAAGTTCATCCCTCCACCCATATTACCACCCTGG TTATTTCCCAAGCCAGCTCCACCCCCTCTACTGTTACCAAACCCAC CCTGATTCCCAAAGCC.

### Human fibroblasts

Fibroblasts were obtained from a 4-mm skin punch biopsy taken following informed consent under local anaesthetic. Biopsies were dissected into ~1-mm pieces and cultured in DMEM, 10% FBS, 1% L-glutamine until fibroblasts grew out from the explants. When confluent, fibroblasts were detached, further expanded and underwent cryopreservation. They were maintained in DMEM with 10% FBS and 1% penicillin/streptomycin under standard culture conditions. Clinical information and genetic changes are summarised in Appendix Table S6.

Transcriptomics analysis methods are described in detail in Appendix Supplementary Materials and Methods.

**Expanded View** for this article is available online.

### Acknowledgements

We thank Gipi Schiavo and Jernej Ule for critical reading of the manuscript. We thank Jernej Ule for his support with the iCLIP experiments. JH thanks Ms Shannon Edwards for her assistance in generating the long intron gene expression plots. PF is funded by an MRC/MNDA LEW Fellowship, the NIHR-UCLH Biomedical Research Centre and the Rosetrees Foundation. AA-A is supported by the Miguel Servet Programme of the ISCiii, Spain (CP15/00153), the UK MRC (MC_UP_A390_1106). This work was partly funded by the Thierry Latran Foundation (PF, AA-A, EMCF, LG), the UK Motor Neuron Disease Association (PF, TR, AA-A, EMCF, LG), the UK Medical Research Council (PF, PS, AU, MS, EMCF), the Rosetrees Foundation (PF, CB, EMCF). National Institute of General Medical Sciences (NIGMS) of the National Institutes of Health (NIH) under Award Number R01GM118530 (to NLF), a starter grant 17-IIP-342 from the ALS Association (to NLF), and an ALS Research Grant from the Judith & Jean Pape Adams Charitable Foundation (to NLF). AEC was supported in part by an NIGMS training grant to the graduate programme in Molecular Biology, Cell Biology and Biochemistry (MCB) at Brown University (T32 GM07601) and a BIBS Graduate Award in Brain Science from the Brown Institute for Brain Science Reisman Fund. KAKENHI nos. 21240043 & 17H00789 (RF & YG), EP and PS are supported by the NIHR BRC GOSH and by Great Ormond Street Children's Charity. MH is funded by the MNDA.

### Author contributions

Conceptualisation: PF, EMCF, AA-A. Methodology: PF, PSi, JH, KL, TR, HO, AEC, YG, ML, LT, FB, LG, EB, VP, EMCF, AA-A. Formal analysis: PF, PSi, JH, KL, TR, HO, JMB-A, BK, YY, ML, AEC, WE, GC, EPa, PSt, EW, DEH, FB, LG, EB, VP, EMCF, AA-A. Investigation: PF, PSi, TR, HO, JMB-A, BK, AU, YY, NB, CB, TC, AEC, AMM, AM-G, MS, JM, SC, WE, GC, MG, RF, YG, EPa, EPe, PSt, MH, ACa, ACh, AMI, EW, EB, AA-A. Writing original draft: PF, AA-A. Writing, review and editing: PF, PSi, JH, KL, TR, HO, EB, AMI, NLF, FB, EB, VP, EMCF, AA-A. Supervision: PF, EB, VP, EMCF, AA-A.

### Conflict of interest

The authors declare that they have no conflict of interest.

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
