## [Review Process File · The EMBO Journal]

Endogenous TDP-43 mutant mice have novel gain of splicing function and ALS characteristics in vivo

Pietro Fratta, Prasanth Sivakumar, Jack Humphrey, Kitty Lo, Thomas Ricketts, Hugo Oliveira, Jose M Brito-Armas, Bernadett Kalmar, Agnieszka Ule, Yichao Yu, Nicol Birsa, Cristian Bodo, Toby Collins, Alexander E. Conicella, Alan Mejia Maza, Alessandro Marrero-Gagliardi, Michelle Stewart, Joffrey Mianne, Silvia Corrochano, Warren Emmett, Gemma Codner, Michael Groves, Ryutaro Fukumura, Yoichi Gondo, Mark Lythgoe, Erwin Pauws, Emma Peskett, Philip Stanier, Lydia Teboul, Martina Hallegger, Andrea Calvo, Adriano Chiò, Adrian M. Isaacs, Nicolas L. Fawzi, Eric Wang, David E. Housman, Francisco Baralle, Linda Greensmith, Emanuele Buratti, Vincent Plagnol, Elizabeth M.C. Fisher and Abraham Acevedo Arozena.

Review timeline:

Submission date:	20 th November 2017
Editorial Decision:	09 th January 2018
Revision received:	15 th February 2018
Editorial Decision:	14 th March 2018
Revision received:	15 th March 2018
Accepted:	22 nd March 2018

Editor: Anne Nielsen

Transaction Report:

1st Editorial Decision

09th January 2018

Thank you for submitting your manuscript for consideration by the EMBO Journal. It has now been seen by three referees whose comments are shown below.

As you will see from the reports, the referees all express interest in the findings reported in your manuscript, although they also raise a number of experimental and conceptual concerns that you will have to address before they can support publication here. Most importantly, you will see that all three referees find that more information is needed on the prevalence and implications of the skiptic splice events seen in the LCD mutant mice. In addition, they ask for a number of additional controls/clarifications to be included and point to a number of sections in the manuscript that should be rephrased or where some conclusions may need to be toned down.

Given the referees' positive recommendations, I would like to invite you to submit a revised version of the manuscript, addressing the comments of all three reviewers. I should add that it is EMBO Journal policy to allow only a single round of revision, and acceptance of your manuscript will therefore depend on the completeness of your responses in this revised version.

REFeree REPORTS

Referee #1:

Pietro Fratta and his collaborators present here a highly interesting and relevant manuscript on TDP-43 function and dysfunction in vivo. Using state of the art molecular and physiological techniques, they convincingly show that a mutation in TDP-43 LCD domain could lead to a so-called splicing "gain of function". As TDP-43 is a key protein involved in amyotrophic lateral sclerosis and fronto-temporal dementia, their findings have broad relevance for these human diseases.

general summary

The investigation of TDP43 function in adult mice has been severely hampered by the lack of relevant animal models. In particular, the use of cDNAs driven by strong promoter led to artifactual phenotypes unrelated to motor neuron disease. Taking this into account, the authors of this manuscript sought to identify point mutations of TDP-43 in a vast ENU-induced mutagenesis program. They identify two new TDP-43 mutant lines in their ENU mouse library. The two mutations are located in two different domains of the protein. The first one, called RRM2mutant, is located in one of the RNA-binding domains of TDP-43 and the authors nicely show that this is a hypomorphic allele. The second one, called LCDmut, carries a mutation in the low complexity domain, a hotspot of ALS-associated mutations. This specific mutation, although currently unknown in patients, seems to lead to similar altered biophysical properties as documented in ALS-related mutants. The authors perform a very comprehensive and exhaustive survey of known TDP-43 functions, including splicing function (using minigene splicing assays), RNA binding assays (using EMSA and iCLIP), gene expression (using RNAseq), and convincingly demonstrate that the RRM2 mutant is a partial loss of function mutant, while the LCD mutant displayed altered splicing function. Strikingly, a new class of splicing events was identified as elicited by TDP43 LCD mutation and involves constitutively included exons that were skipped in LCD mutant mice. The authors then use bioinformatic analysis to suggest that these changes could be linked to direct TDP-43 action on these exons. Last, the authors show that LCDmut mice show mild neuromuscular phenotypes in a manner strikingly similar to what has been recently shown by the same group and others in FUS knock in mice.

The methodologies used are outstanding and conclusions reached are extremely solid. The writing of the paper is clear, and figures are of great quality.

In all, this is an outstanding paper, with findings that could be critical in our understanding of TDP43 dysfunction in ALS and FTD.

Major concerns

Three major concerns could be addressed to strengthen the results.

1) What is the extent of the skiptic exons related events ?

The authors identify 44 skiptic exons out of 523 differentially spliced exons. This could be significant in terms of disease pathogenesis yet it is difficult to understand with the current manuscript. Are the >90% remaining differentially expressed exons potential skiptic or almost skiptic exons that were not identified due to very stringent bioinformatic analysis criteria ? Among the known TDP-43 target exons, what is the proportion of skiptic exons ? The text mentions that skiptic exons show TDP43 binding, yet are there other features that could explain why only a fraction of constitutively expressed exons binding TDP43 become skipped ? Also the text mentions that Fig4B shows the conservation of SEs between species, and this is not the case. In all, the section characterizing skiptic exons would benefit to be expanded and better explained.

2) Are skiptic exons present in human ALS ?

If the authors would be able to provide evidence that similar skiptic exons exist in ALS would be a major addition that could potentially largely increase the relevance of the study. This might be possible using existing datasets of RNAseq performed in sporadic or C9ORF72 patients or iPS derived cells. If it is not technically possible using existing datasets (for instance due to lack of depth of sequencing, or other technical reasons), this should be stated in the discussion and remains an obvious extension of their current work. Importantly also, it would be of tremendous importance to determine whether the skiptic exons are found in the same genes in humans and mice (as it is not the case for cryptic exons).

3) Do LCDmut mice display motor neuron degeneration ?

The authors provide interesting data on the existence of a neuromuscular phenotype, associated with the development of p62 pathology, yet they do not show motor neuron numbers or neuromuscular junction histology. This would definitely strengthen the relevance to ALS to show that motor

neurons are (partially) lost. Also, is p62 pathology in motor neurons? Last, the authors do not comment whether they aged similarly RRM2mut mice. If this was the case, did they notice similar motor neuron defect ?

Minor concerns

- 1) The authors should be careful in their conclusion that skiptic exons are induced by direct TDP43 binding or lead to downregulation as they do not provide any direct evidence for this. The evidence in bioinformatic analysis is extremely interesting but remains correlative.
- 2) EMSA of figure S3 is not of very good quality. Could the authors provide a better image showing the dose response ?
- 3) Blots of supplementary Figure 4 are likely flipped : they show an opposite result as compared with the quantification...
- 4) The discussion is the weakest part of this manuscript, and remains extremely factual. It would be very valuable to discuss the relevance of the study to ALS/FTD, and also its limitations (eg that the exact same mutation has not been observed in patients). This is not appropriately done in the current manuscript.

Non-essential suggestions

In most panels showing electrophoresis gels, the legend is indicated below the gel (and below the histograms) (see for instance Figure 1B or EV2C). This is not obvious for the reader.

Referee #2:

Fratta et al. identified two mouse line with ENU induced mutation in Tardbp/TDP-43. They show that a mutation in the RRM2 domain causes loss of function, while a mutation in the low complexity domain (LCD, where most ALS-causing mutations reside) may cause a gain of function phenotype on splicing level. Analyzing mutation at endogenous expression levels is advantageous compared to transgenic models, but of course neither of the mutations is known to cause ALS in humans. The authors focus on splicing and expression effects in mouse embryonic fibroblasts and mouse tissue. The LCD mutation is associated with loss of constitutive exons, which they termed "skiptic exons" to contrast the "cryptic exons" appearing in TDP-43 LOF tissue. They argue skiptic exons represent TDP-43 GOF toxicity. This is a very interesting hypothesis that requires further experimental support. Especially the nature of the putative GOF mechanism and its role in ALS needs to be investigated further.

Specific comments:

- Indicate whether heterozygous or homozygous animals/cells were used in all figures or legends and mention the mouse strain. EMBO mandated statistical information is also missing in most legends.
- Fig S1: Show images of actual droplet formation due to phase separation. Is full length TDP-43 used? It seems that most publications use only C-terminal fragments. Q331K was reported to enhance aggregation in a similar assay (Johnson et al., JBC 2009) and inhibit phase separation for a C-terminal fragment (Conicella et al, Structure 2017).
- Fig 1E,F: Mere analysis of the binding motif does not make full use of the dataset. Provide more data on, e.g. correlation of CLIP efficacy at gene/exon level. Show coverage at TDP-43 autoregulation site, CFTR ex9 and SORT1 ex18 for the different TDP43 variants.
- Fig S4: provide actual sequence of scrambled shRNA.
- Fig S4C/D: Are the images flipped? It seems that Tardbp shRNA upregulated TDP-43 levels.
- Fig 2A: label affected exons within the figure and provide expression data in supplemental material. Were cryptic exons identified in loss of function conditions?
- Fig 2B: Adding the percentages for the different genotypes results in far less than 100%. What genotype are the remaining 20-50% of mice?
- Fig 3: To confirm that skiptic exons are due to a general GOF and not due to this specific mutation the authors should quantify skiptic exons in TDP-43 overexpression conditions and ideally with other C-terminal mutations. Several mouse models are available and maybe cell culture would be informative too.
- Fig 4: panels are referenced wrong in text
- Fig 5: Is there visible neuron loss?
- Fig 5/S5: TDP-43-negative p62-positive inclusions are not common in sporadic ALS or in cases

with TDP-43 mutation. Define these aggregates better. Are the Ub/P62-positive inclusions restricted to the spinal cord. Could this be due to TDP-43 mediated defects in autophagy (Xia et al, EMBO J 2016)? Analyse TFEB and autophagy in these mice. This could help defining whether LCDmut mice are mainly GOF or LOF.

-Page 9 "... suggesting TDP43 GOF plays a key role in driving TDP-43-related ALS" and the last paragraph of discussion are gross overstatements. Clearly both TDP43 GOF and LOF can be neurotoxic, but the manuscript contains no evidence for actual TDP43 GOF in ALS. Are skiptic exons enhanced in ALS spinal cord?

-Fig EV3: The strange findings on TDP-43 autoregulation need further investigation. Classically, enhanced Tardbp mRNA would argue for LOF-induced autoregulation. Maybe in-depth analysis of the transcripts and CLIP data for the different TDP43 mutants would help. Overexpression expression experiments in vitro could also be useful.

Referee #3:

General summary and opinion about the principle significance of the study, its questions and findings: The work of Fratta et al is an interesting approach for characterization of new mouse alleles for TDP-43 and its capacity in controlling splicing. Because of the relevance of TDP-43 for the pathogenesis of ALS /FTD, the models studies may shed light on disease. The values of the work for the field are in (1) novel mouse mutants carrying point mutations in endogenous Tardbp. (2) TDP-43 bioinformatics and splicing function. (3) exploration of gain- and loss-of TDP-43 function. It is an important piece.

There are several major concerns that should be addressed in revisions:

(I) Exonic Skipping

The authors should better not propose that they observe a new splicing mechanism, as the support and mechanistic insight for it is very limited in this work with a decrease in constitutive exon inclusion being a rare event (48 events in 233,785 human exons).

(I 1) The authors should rule out that these are events depicted at random under the null hypothesis, because lack of correction to multiple hypothesis (233,785 human exons). Accordingly, if the SEs found in LCDmut are effectively picked at random, it may explain why they were not identified in the RRM2mut (~mirror image) model. The data and interpretation of the 33 cryptic exons may suffer from a similar problem.

(I 2) The most likely mechanistic explanation is some drop in the performance/fidelity of the spliceosome that harbors a mutant TDP-43 protein. Percent spliced in (PSI) of constitutive exons under wild type conditions is defined as >0.95 . Therefore, the authors expect inevitable and infrequent mistakes of the splice some at up to 5% even under wild-type conditions. Does deltaPSI of 0.1 means that non-canonical splicing (exon skipping) is seen at PSI of 0.95 (normal performance) minus 0.1 (delta PSI) = coming to $PSI < 0.85$? What is the real PSI for all 48 skipped exons? What is the median PSI for the cohort of 48? A value too close to 0.95 in any of these genes, decreases the confidence in the interpretation as it approximates normal fidelity. A mutant median PSI that is changing > -0.3 (new PSI of 0.65) is reassuring. The offered value of -0.3PSI is arbitrary of course but there should be a way to extend to the reader that there is a significant change. Revisiting these data might change the way the referee and reader look on the proposed skipping event.

(I 3) Discuss a potential mechanism / model for skipping exons and how molecularly, it is different from just insufficiency.

(II) Rewrite text and reduced over interpretation. Text tends to be polemic and winding. A simple linear explanations will help the readers. Play down interpretations.

(III) Additional points:

(III 1) Usage of the minigene-splicing reporter (from Buratti) in Fig. 1B. Authors claim that the vectors works as a "well documented" positive control. They should provide the data.

(III 2) Figure 1C Will be nicer to see a representative gel image of PCR product.

(III 3) Fig 2B Genetics interactions: what are the absolute numbers counted dead and viable animals in liters.

(III 4) Compound heterozygote alleles exhibit 6.5% survival instead of the expected 25%. It is harsh lethality, and although more viable than each mutant on its own, the authors should increase the confidence in the data but showing the numbers and also discuss the incomplete rescue.

(III 5) Sashimi plots: improve presentation and associated explanations: use same graphic system. Add Y axis. Explain in details to reader can appreciate differences between experimental conditions (Sashimi plot in e.g. Fig 3). Consider a clear way to present the data (e.g. including counts as in Figure 2C). It is not clear how numbers and inclusion percentile are calculated for sashimi plot in Figure 2C. Explain how counts are calculated into inclusion percentage. Similar numeric representations should be available in all sashimi plots.

(III 6) Define LOF and GOF:

Is the benchmark for loss of function shRNA knockdown of TDP-43? Might there be another reference benchmark?

(III 7) Fig. 2A are the lists correlated with rigorous by statistical method (e.g. Pearson correlation)? Supp Figure 5 IHC. Add positive control tissue.

(III 8) Figure legends: what statistic test used?

(III 9) Presenting data as MA / Bland-Altman plot for visualizing of data that is presented now only as volcano plots will allow grasping the differences between measurements while retaining the average mean expression level that is not presented on volcano plot.

(IV) Minor comments:

Textual:

page 5 second paragraph: Figure 1A should be 1B

page 8 upper paragraph Figure 4C should be 4D?

page 8 upper paragraph Figure 4D should be 4E?

Fig 5 legend 7 months, vs 9 month in panel 5A?

Legend fig 3: correct text: "spliced exons that have either of"

Page 10 : GOF is sufficient to initiate the neurodegenerative process" is it true?

Referee #1:

Pietro Fratta and his collaborators present here a highly interesting and relevant manuscript on TDP-43 function and dysfunction in vivo... their findings have broad relevance for these human diseases...The methodologies used are outstanding and conclusions reached are extremely solid. The writing of the paper is clear, and figures are of great quality. In all, this is an outstanding paper, with findings that could be critical in our understanding of TDP43 dysfunction in ALS and FTD.

Major concerns

Three major concerns could be addressed to strengthen the results.

1) What is the extent of the skiptic exons related events?

The authors identify 44 skiptic exons out of 523 differentially spliced exons. This could be significant in terms of disease pathogenesis yet it is difficult to understand with the current manuscript. Are the >90% remaining differentially expressed exons potential skiptic or almost skiptic exons that were not identified due to very stringent bioinformatic analysis criteria?

- To identify the skiptic exon related events, we have indeed used very stringent p-values after multiple testing correction (FDR<0.05) throughout the paper so that the status of the skiptic exons we show is without ambiguity. The 90% of other alternatively spliced exons are simply exons that are differentially regulated in the presence of mutant TDP-43, but are even normally present in two forms (i.e. they are present as spliced and un-spliced in WT), whilst the skiptic exons are those that are not skipped in WT.

Among the known TDP-43 target exons, what is the proportion of skiptic exons?

The text mentions that skiptic exons show TDP43 binding, yet are there other features that could explain why only a fraction of constitutively expressed exons binding TDP43 become skipped?

- We thank the referee for this comment which has highlighted an interesting point. We have now added a table (Supplementary Table 5) illustrating that, whilst TDP-43 RNA binding clearly is increased in differentially spliced exons (18.8%) and even more in skiptic exons (66.0%) compared to all exons (5%), there are numerous constitutive exons that have TDP-43 binding and do not become skiptic in TDP-43 GOF.

This is likely because the splicing of pre mRNAs is a complex mechanism involving multiple factors including spliceosome components and RNA binding proteins such as TDP-43 and other hnRNPs. Further studies will allow us to better understand why in some events TDP-43 carries more "weight" than in others, but such studies are well beyond the scope of this paper. We have now added the following sentence in the discussion "this is likely due to the cooperative nature of splicing and the fact that TDP-43 is only one of many factors involved in the regulation of the inclusion/exclusion of the hundreds of exons it binds to".

Also the text mentions that Fig4B shows the conservation of SEs between species, and this is not the case.

- The manuscript should have read **Fig 4C** and we have now made this correction.

In all, the section characterizing skiptic exons would benefit to be expanded and better explained.

- In light of this and the other referees' comments, we have expanded the explanation of skiptic exons both in the results and the discussion sections.

2) Are skiptic exons present in human ALS ?

If the authors would be able to provide evidence that similar skiptic exons exist in ALS would be a major addition that could potentially largely increase the relevance of the study. This might be possible using existing datasets of RNAseq performed in sporadic or C9ORF72 patients or iPS derived cells. If it is not technically possible using existing datasets (for instance due to lack of depth of sequencing, or other technical reasons), this should be stated in the discussion and remains an obvious extension of their current work. Importantly also, it would be of tremendous importance to determine whether the skiptic exons are found in the same genes in humans and mice (as it is not the case for cryptic exons).

- We agree that this is a very important issue, but there are no publicly available RNA-seq datasets from human tissue/cells carrying *TARDBP* mutations. We do not believe that sporadic or C9orf72 iPSC derived MNs, where no *TARDBP* mutations are present and no TDP-43 mislocalisation is observed, are relevant to our current findings.
- However, to address this point we have been able to produce two important new analyses:
 - (1) we have analysed human ALS fibroblasts carrying *TARDBP* mutations and have now validated via RT-PCR of this patient-derived material, the presence of two skiptic exons identified in the *LCDmut* mice. This new data is now shown in Figure 7.
 - (2) we have also used a novel knock-in mouse model that carries the disease-causative Q331K mutation to validate skiptic exons and splicing GOF (both using the CFTR minigene assay and validating GOF events on endogenous RNAs by RT-PCR). This new data is shown in Figure 7.

3) Do LCDmut mice display motor neuron degeneration?

The authors provide interesting data on the existence of a neuromuscular phenotype, associated with the development of p62 pathology, yet they do not show motor neuron numbers or neuromuscular junction histology. This would definitely strengthen the relevance to ALS to show that motor neurons are (partially) lost.

- We agree that motor neuron dysfunction/death is a key issue. Thus we now have performed and include motor neuron counts from the lumbar spinal cord of *LCDmut* mice, showing there is a significant reduction in MN numbers (28% reduction; $p=0.0159$). We are including this new data in Figure 6D. These findings highlight the validity and utility of our new mouse model to give new insight into potential molecular mechanisms of motor neuron dysfunction and death.

Also, is p62 pathology in motor neurons? Last, the authors do not comment whether they aged similarly *RRM2mut* mice. If this was the case, did they notice similar motor neuron defect?

- Regarding p62 pathology, we have expanded our analysis to the brain and show that no difference between *LCDmut* and WT occur in hippocampus, but in brainstem motor nuclei p62 inclusions are present, similarly to *LCDmut* spinal cord. We have added the following statement as to relation with motor neurons: *"We investigated whether this phenomenon was widespread in the CNS, and brain immunohistochemistry showed an increase in p62 pathology in LCDmut brainstem motor nuclei, whilst no change was detectable in other brain regions including the hippocampus (Figure S8). Intriguingly, although p62 inclusions are preferentially enriched in areas where MN are located (spinal cord ventral horns and brainstem motor nuclei), they do not appear to be localised to the cytoplasm of MN cell bodies."*
- We have similarly aged heterozygous *RRM2mut* mice (homozygous mutants are not viable) and while these mice are smaller than wildtype littermates they do not show any obvious phenotypes

in our tests of motor function such as grip-strength, and do not show p62 pathology. A sentence has been added to the text to highlight this.

Minor concerns

1) The authors should be careful in their conclusion that skiptic exons are induced by direct TDP43 binding or lead to downregulation as they do not provide any direct evidence for this. The evidence in bioinformatic analysis is extremely interesting but remains correlative.

- We agree and have changed the results and discussion section as shown in the text. One relevant example of change is the heading of one of the Results paragraphs: "~~Skiptic exons are induced by~~ linked to direct TDP-43 RNA binding"

2) EMSA of figure S3 is not of very good quality. Could the authors provide a better image showing the dose response?

- We have added to the EMSA in **Supplementary Figure 4**, a more quantitative EMSA to assess if the M323K mutation affects the ability of TDP-43 to bind RNA. We used an increasing amount of "hot" RNA carrying a UG₆ repeat known to bind TDP43 avidly. We used a fixed amount of recombinant wildtype or M323K mutant TDP-43 protein against the UG₆ repeats. The new results show, as previously, that at least to the limits of the EMSA technique used, there are no significant differences between the binding of wildtype and *LCDmut* TDP43.

3) Blots of supplementary Figure 4 are likely flipped: they show an opposite result as compared with the quantification...

- This was detected by Referee 1 and 2, and we apologise for this error, which we have corrected. The figure is now called Figure S5.

4) The discussion is the weakest part of this manuscript, and remains extremely factual. It would be very valuable to discuss the relevance of the study to ALS/FTD, and also its limitations (eg that the exact same mutation has not been observed in patients). This is not appropriately done in the current manuscript.

- We welcome the opportunity to provide a more in depth Discussion that focuses on disease issues. We have amended the Discussion accordingly and include a new a new paragraph on the value and limitations of our study to ALS/FTD, as shown in the text.

Non-essential suggestions

In most panels showing electrophoresis gels, the legend is indicated below the gel (and below the histograms) (see for instance Figure 1B or EV2C). This is not obvious for the reader.

- We are now indicating the legend also on the top of the gels.

Referee #2:

Fratta et al. identified two mouse lines with ENU induced mutation in *Tardbp*/TDP-43. They show that a mutation in the RRM2 domain causes loss of function, while a mutation in the low complexity domain (LCD, where most ALS-causing mutations reside) may cause a gain of function phenotype on splicing level. Analyzing mutation at endogenous expression levels is advantageous compared to transgenic models, but of course neither of the mutations is known to cause ALS in humans. The authors focus on splicing and expression effects in mouse embryonic fibroblasts and

mouse tissue. The LCD mutation is associated with loss of constitutive exons, which they termed "skiptic exons" to contrast the "cryptic exons" appearing in TDP-43 LOF tissue. They argue skiptic exons represent TDP-43 GOF toxicity. This is a very interesting hypothesis that requires further experimental support. Especially the nature of the putative GOF mechanism and its role in ALS needs to be investigated further.

Specific comments:

-Indicate whether heterozygous or homozygous animals/cells were used in all figures or legends and mention the mouse strain. EMBO mandated statistical information is also missing in most legends.

- We now include all relevant information in all figures legends, including the EMBO mandated statistical information.

-Fig S1: Show images of actual droplet formation due to phase separation. Is full length TDP-43 used? It seems that most publications use only C-terminal fragments. Q331K was reported to enhance aggregation in a similar assay (Johnson et al., JBC 2009) and inhibit phase separation for a C-terminal fragment (Conicella et al, Structure 2017).

- As the referee suggested, C-terminal fragments were used for the analysis and we are making this information more apparent in the figure legend. As previously reported for other C-terminal mutations (e.g. Conicella *et al*, Structure 2017), the C-terminal TDP43 fragment containing the M323K mutation behaves as the Q331K control used, inhibiting C-terminal phase separation.

-Fig 1E,F: Mere analysis of the binding motif does not make full use of the dataset. Provide more data on, e.g. correlation of CLIP efficacy at gene/exon level. Show coverage at TDP-43 autoregulation site, CFTR ex9 and SORT1 ex18 for the different TDP43 variants.

- We have now added iCLIP peaks to sashimi plots showing splicing events in *Sort1*, *Eif4h* and *Tardbp* in EV1 and Supplementary figure 9. We have not done this for CFTR, as this is a minigene that was used in primary cell experiments (MEFs). CFTR is not expressed in brain and spinal cord – the tissues where iCLIP was performed. Lastly, unfortunately iCLIP is not a quantitative technique, and with our current results we are unable to comment on binding efficacy.

-Fig S4: provide actual sequence of scrambled shRNA.

- The sequence is now provided in the supplementary materials. This figure is now Figure S5.

-Fig S4C/D: Are the images flipped? It seems that *Tardbp* shRNA upregulated TDP-43 levels.

- As above, we apologise for this error which has been corrected in the manuscript.

-Fig 2A: label affected exons within the figure and provide expression data in supplemental material. Were cryptic exons identified in loss of function conditions?

- We have now added a further table (Supplementary Table 2) where exons from Figure 2A are identified with name, genomic coordinates, fold change and significance.
- We do identify cryptic exons in F210I and TDP-43 KD datasets, as correctly predicted by the referee. We include in the rebuttal an example of this below, where the cryptic exon in *Adnp2* is present only in RRM2mut and TDP-shRNA, whilst absent in LCDmut and in all controls. We do not comment

about cryptic exons at this point in the manuscript as the cryptic exon analysis is of higher quality and relevance on the high-depth RNA-seq performed on CNS and presented in Figure EV2.

-Fig 2B: Adding the percentages for the different genotypes results in far less than 100%. What genotype are the remaining 20-50% of mice?

- We apologise for the typographical errors in this figure and which we have corrected, we are also including the number of mice per genotype.

-Fig 3: To confirm that skiptic exons are due to a general GOF and not due to this specific mutation the authors should quantify skiptic exons in TDP-43 overexpression conditions and ideally with other C-terminal mutations. Several mouse models are available and maybe cell culture would be informative too.

- This is a critical point raised by the reviewer, and indeed it is important to show that the splicing GOF is not a peculiar effect of one single ENU-induced TDP-43 mutation. We believe that in overexpression models, it is difficult to identify the effects of pathogenic mutation because of the major impact of TDP-43 overexpression *per se* on splicing, as identified for example, in Arnold *et al* (PNAS, 2013) – hence our investment in studying the physiological models presented in our paper. However, to address the referee’s point we now include data from our novel, unpublished, *Tardbp* Knock In mice that carry a pathogenic *human* TDP43 Q331K mutation, expressed at endogenous levels (as in *LCDmut* and *RRM2mut*). We present data from MEFs and spinal cord of 3 month old Q331K mice in Figure 7. However, the overall characterisation of this novel mouse strain is still ongoing and will be published in the future.

Our new data from Q331K mice validate the splicing gain of function identified in the *LCDmut* mice. We show the Q331K mutation leads to a splicing gain of function in MEFs, as exemplified by the CFTR minigene assay and by the splicing of a number of TDP43 endogenous targets (*Sortilin*, *Eif4h* and *Kcnp2*) analysed via RT-PCR. These new data are presented in Figure 7. Moreover, we were also able to identify skiptic exons in Q331K mice, as presented in Figure 7, further confirming that

the effect described in our paper is not caused only by the *LCDmut*, but also by ALS-causing C-terminal TDP-43 mutants.

We believe, that these new data, together with the new data on skiptic exons produced from ALS patient-derived fibroblasts (Figure 7), show that indeed C-terminal mutations in TDP-43 lead to a splicing gain of function that is also present in the disease state in human and mouse.

-Fig 4: panels are referenced wrong in text

- Again, we apologise for this error and have corrected the text.

-Fig 5: Is there visible neuron loss?

- This is a very important point and we now present quantifications of MNs showing MN loss. As above (response to referee 1), we now include counts from the sciatic pool of *LCDmut* mice showing statistically significant motor neuron loss, please see new Figure 6D.

-Fig 5/S5: TDP-43-negative p62-positive inclusions are not common in sporadic ALS or in cases with TDP-43 mutation. Define these aggregates better. Are the Ub/P62-positive inclusions restricted to the spinal cord. Could this be due to TDP-43 mediated defects in autophagy (Xia et al, EMBO J 2016)? Analyse TFEB and autophagy in these mice. This could help defining whether *LCDmut* mice are mainly GOF or LOF.

- We have now added staining data from mouse brain that illustrate these inclusions are not diffusely present throughout the CNS: we see no changes in hippocampus, but, similarly to what observed in the spinal cord ventral horn, we observe strong p62 inclusions in the bulbar motor nuclei in *LCDmut* mice. These data, along with the preferential presence in ventral horns supports the relevance of these inclusions to the motor degeneration. Whether the lack of TDP-43 in these inclusions is due to mouse/human differences needs to be further explored.
- In terms of possible defects in autophagy, it is clearly an interesting possibility. However, the methods to assess autophagic flux *in vivo* are notoriously difficult. Moreover, in the case of *LCDmut* mice, we are only detecting inclusion pathology sparsely in restricted CNS areas critical for ALS: lumbar spinal cord and brainstem motor nuclei. Therefore, analysis of autophagic markers by biochemistry approaches would likely be confounded by the presence of "affected" and "non-affected" areas/cells. Despite this, to assess if *LCDmut* affects the expression of TFEB or target genes, we analysed the RNA-seq data from adult *LCDmut* spinal cord, but could not find any significant differences in expression for *Tfeb* itself or in a selection of its autophagy target genes: *Atg9b*, *Uvrag*, *Becn1* or *Gabarap*. Moreover, in contrast to the data from Xia et al focussing on the effects of TDP-43 loss of function on autophagy, *Raptor* expression is not changed in *LCDmut* spinal cord.

-Page 9 "... suggesting TDP43 GOF plays a key role in driving TDP-43-related ALS" and the last paragraph of discussion are gross overstatements. Clearly both TDP43 GOF and LOF can be neurotoxic, but the manuscript contains no evidence for actual TDP43 GOF in ALS. Are skiptic exons enhanced in ALS spinal cord?

- This is indeed a key point. To address this criticism and comments above from Referee 1 we now present important new data from Q331K mutant mice (i.e. human pathogenic mutation) and from fibroblasts of ALS patients with TDP-43 mutations that shows skiptic exons. However, in light of all of the reviewers' comments, we have changed the discussion accordingly and have replaced the above statement. This now reads: "Finally, we show that, at least in this

model, LOF is not necessary for initiating the neurodegenerative processes, whilst highlighting that the occurrence of splicing GOF is associated with the early stages of disease”.

-Fig EV3: The strange findings on TDP-43 autoregulation need further investigation. Classically, enhanced *Tardbp* mRNA would argue for LOF-induced autoregulation. Maybe in-depth analysis of the transcripts and CLIP data for the different TDP43 mutants would help. Overexpression expression experiments in vitro could also be useful.

- We agree with the referee that the findings on TDP43 autoregulation are unexpected. It is important for readers to be able to see and analyse our data, and therefore we now include a zoom in of the RNA-seq traces and the CLIP data focussing on the 3' of the *Tardbp* locus from *LCDmut* and *RRM2mut* in Supplementary Figure 9.

Referee #3:

The work of Fratta et al is an interesting approach for characterization of new mouse alleles for TDP-43 and its capacity in controlling splicing. Because of the relevance of TDP-43 for the pathogenesis of ALS /FTD, the models studies may shed light on disease. The values of the work for the field are in (1) novel mouse mutants carrying point mutations in endogenous *Tardbp*. (2) TDP-43 bioinformatics and splicing function. (3) exploration of gain- and loss-of TDP-43 function. It is an important piece.

There are several major concerns that should be addressed in revisions:

(I) Exonic Skipping

The authors should better not propose that they observe a new splicing mechanism, as the support and mechanistic insight for it is very limited in this work with a decrease in constitutive exon inclusion being a rare event (48 events in 233,785 human exons).

- It was not our intention to suggest that we have uncovered is a new splicing mechanism, and in light of Referee 3's comment we have carefully reviewed and modified our paper to check there is no suggestion of this. We have also added the following statement (amongst other highlighted in text) to the discussion to avoid ambiguity: *“Thus CEs and SEs are not a novel splicing mechanism, but the effect of LOF and GOF respectively on a subset of constitutively included and excluded exons”*.

(I 1) The authors should rule out that these are events depicted at random under the null hypothesis, because lack of correction to multiple hypothesis (233,785 human exons). Accordingly, if the SEs found in *LCDmut* are effectively picked at random, it may explain why they were not identified in the *RRM2mut* (~mirror image) model. The data and interpretation of the 33 cryptic exons may suffer from a similar problem.

- We thank the reviewer for the suggestion. We agree that it is important to rule out we are just detecting events due to random sampling. First, all our analysis and significance thresholds are selected and done with extremely thorough multiple testing statistical corrections outlined in the methods. Secondly, to further address the point, we have performed our analysis on 50 random sample permutations and results are now added in Supplementary Table 4. As can be seen, out of the 50 permutations 45 identified 0 skiptic exons, 4 identified 1 and 1 identified 2. This analysis does further demonstrate the specificity of our analysis and the fact that these are not picked at random. Moreover, the fact that we were able to validate some skiptic exons in Q331K mice and patient-derived fibroblasts further supports the specificity for the identified skiptic exons.

(I 2) The most likely mechanistic explanation is some drop in the performance/fidelity of the spliceosome that harbors a mutant TDP-43 protein. Percent spliced in (PSI) of constitutive exons under wild type conditions is defined as >0.95 . Therefore, the authors expect inevitable and infrequent mistakes of the spliceosome at up to 5% even under wild-type conditions. Does deltaPSI of 0.1 means that non-canonical splicing (exon skipping) is seen at PSI of 0.95 (normal performance) minus 0.1 (delta PSI) = coming to $PSI < 0.85$? What is the real PSI for all 48 skipped exons? What is the median PSI for the cohort of 48? A value too close to 0.95 in any of these genes, decreases the confidence in the interpretation as it approximates normal fidelity. A mutant median PSI that is changing > -0.3 (new PSI of 0.65) is reassuring. The offered value of -0.3 PSI is arbitrary of course but there should be a way to extend to the reader that there is a significant change. Revisiting these data might change the way the referee and reader look on the proposed skipping event.

- Again, the reviewer highlights a critical point. We have now changed the text to make the criteria for defining the skipped exons more clear. These are exons that have a $PSI > 0.95$ in WT, with a change in $PSI > 0.05$. This means that the minimal splicing change to be classified as SEs would a change from 0.949 to 0.90 that passes multiple testing correction ($FDR < 0.05$).
- To give the reviewer and readers a better idea of the entity of changes we have significantly modified Supplementary Table 3. We have included for all SEs the values of WT PSI, *LCDmut* PSI, DeltaPSI, and fold change. We also report the mean and median for the identified 47 SEs: in mean SEs shift from a PSI of 0.98 (WT) to 0.86 (*LCDmut*). The median fold increase in splicing exclusion is 8.6 fold. Interestingly, the changes are similar to those found in all other cassette splicing changes found in the dataset: mean deltaPSI 0.11, median deltaPSI 0.9, confirming the entity of changes of SEs makes them legitimate splicing changes.

(I 3) Discuss a potential mechanism / model for skipping exons and how molecularly, it is different from just insufficiency.

- We thank Referee 3 for this opportunity and we have added text in discussion accordingly. In summary, we present data that links the exon skipping GOF with TDP-43 RNA binding sites suggesting that the GOF is likely mediated by TDP-43 binding, but the mechanism needs further understanding. We discuss the potential impact of TDP-43 C-terminal mutations on protein-protein interactions and how further studies will be needed to investigate this.
- We bring significant evidence to show that the GOF exon skipping is different from insufficiency and we summarise this in Figure 4. The evidence supporting the differences with haploinsufficiency are multiple, primarily based on the finding that *LCDmut* causes opposite changes compared to insufficiency, including:
 - The CFTR minigene (Figure 1)
 - The splicing of endogenous targets by RT-PCR (Figure 1)
 - The comparison of MEF RNA-seq from *RRM2mut*, *LCDmut* and TDP-43 KD.
 - The intercrosses between *LCDmut* and *RRM2mut* mice.
 - The association of TDP-43 binding motifs to opposite splicing events (EV1)
 - The finding of CEs and SEs (Figure 3; EV2) and their association with RNA binding by CLIP data (Figure 5).

(II) Rewrite text and reduced over interpretation. Text tends to be polemic and winding. A simple linear explanations will help the readers. Play down interpretations.

- We hope have now addressed the referee's comments as requested.

(III) Additional points:

(III 1) Usage of the minigene-splicing reporter (from Buratti) in Fig. 1B. Authors claim that the vectors works as a "well documented" positive control. They should provide the data.

- We are now including in Supplementary Figure 2, data showing that in our hands the minigene construct is a reporter for TDP43 splicing function, as published in Buratti and Baralle 2001. Since this first paper, the construct has been used by multiple authors to assess TDP43 splicing function.

(III 2) Figure 1C Will be nicer to see a representative gel image of PCR product.

- We thank the referee for this comment and now include representative images in the figure.

(III 3) Fig 2B Genetics interactions: what are the absolute numbers counted dead and viable animals in liters.

- As requested we include the absolute numbers in Figure 2B.

(III 4) Compound heterozygote alleles exhibit 6.5% survival instead of the expected 25%. It is harsh lethality, and although more viable than each mutant on its own, the authors should increase the confidence in the data but showing the numbers and also discuss the incomplete rescue.

- We have now corrected the table and added the absolute numbers. The survival of the compound heterozygotes is 8.8%, which we agree with the reviewer is far from the expected Mendelian 25%. We highlight this now in results and discussion and we discuss the fact that the partial rescue could be due to the fact that, although the two lines show opposite changes due to LOF/GOF, these are not necessarily of the same entity and therefore result in an incomplete rescue of the lethality phenotype.

(III 5) Sashimi plots: improve presentation and associated explanations: use same graphic system. Add Y axis. Explain in details to reader can appreciate differences between experimental conditions (Sashimi plot in e.g. Fig 3). Consider a clear way to present the data (e.g. including counts as in Figure 2C). It is not clear how numbers and inclusion percentile are calculated for sashimi plot in Figure 2C. Explain how counts are calculated into inclusion percentage. Similar numeric representations should be available in all sashimi plots.

- We have now harmonised the presentation of the sashimi plots and included the junction read numbers. The Y axis is dependent on the sample in RNA-seq data and we have therefore not included this – the main purpose of these representations is for the reader to “see” the splicing changes.
- The PSI values have now been included as in the rest of the paper using SG-seq (see methods).

(III 6) Define LOF and GOF:

Is the benchmark for loss of function shRNA knockdown of TDP-43? Might there be another reference benchmark?

- Conventionally the benchmark for LOF is gene KO. As we explain in the manuscript, unfortunately TDP-43 KO does not allow the generation of null mice as this causes a lethality before embryonic day 6 and heterozygous KOs have normal TDP43 protein levels because of auto-regulation. We cannot therefore use a mouse as the LOF benchmark and use TDP-43 knock-down, which has been widely used to study TDP-43 LOF.

(III 7) Fig. 2A are the lists correlated with rigorous by statistical method (e.g. Pearson correlation)?

- In Figure 2A all exons from the mouse transcriptome are plotted. We know that, although TDP-43 is involved in splicing of many exons, the large majority of exons are not regulated by its function. We therefore are not interested in the correlation between the datasets, but we analyse the coherence of changes where an exon is significantly changed in both datasets from the 2-way comparison. The results support LOF for *RRM2mut* and GOF for *LCDmut* along with other data in Figure 1,2,3,5 EV1 and EV2.

(III 8) Figure legends: what statistic test used?

- We now report the EMBO mandatory statistics data on each figure legend and we apologise for not doing this in the original submission.

(III 9) Presenting data as MA / Bland-Altman plot for visualizing of data that is presented now only as volcano plots will allow grasping the differences between measurements while retaining the average mean expression level that is not presented on volcano plot.

- We have included MA plots in supplementary data, as requested (Supplementary Figure 5).

Supp Figure 5 IHC. Add positive control tissue.

- We have now added this.

(IV) Minor comments:

Textual:

page 5 second paragraph: Figure 1A should be 1B

page 8 upper paragraph Figure 4C should be 4D?

page 8 upper paragraph Figure 4D should be 4E?

Fig 5 legend 7 months, vs 9 month in panel 5A?

Legend fig 3: correct text: "spliced exons that have either of"

- We thank the reviewer for identifying the errors, which are all now corrected.

Page 10 : GOF is sufficient to initiate the neurodegenerative process" is it true?

We have accordingly toned down the discussion and the last paragraph. We conclude that LOF is dispensable for initiation of neurodegeneration, and that GOF is occurring. We have therefore toned down the claim of causality, as the referee asked.

As you will see, the referees are overall satisfied with the revised manuscript and support publication here. However, ref #2 has a few reservations with the new data that has been included and asks that you improve/clarify the data presentation to make the effects comparable between different figures. I would therefore invite you to submit a final version of the manuscript in which you incorporate the remaining changes requested by ref #2.

REFEREE REPORTS

Referee #1:

The authors adequately answer to my previous comments.

Referee #2:

Overall the manuscript improved and now contains a thorough characterization of the two TDP-43 ENU mice. The question how relevant these effects are for ALS is not fully answered. The data from a mouse line with a known pathogenic TDP-43 mutation (Q331K) and patient fibroblasts in Fig. 7 is not very convincing. I was puzzled by the apparent discrepancy of the large effects in the quantification and the virtual absence of effects in the gel pictures (especially Fig. 7C/D - compare Fig 3C) until I noticed that the y-axis in most graphs is not starting at 0. For example, for Pacrgl the exon inclusion merely drops from 96% to 91%. This may be statistically significant but is not highly convincing. All y-axes in Fig 7 should start at 0 and the interpretation has to be more cautious. In Fig 7E, quantification for all 7 SEs tested in the patient fibroblasts should be shown using the same presentation/calculation as in the other figures (PSI instead of normalized skiptic exons). I assume these effects will also appear much smaller then. The current presentation in Fig 7 greatly exaggerates how wide-spread exon skipping occurs in bona fide ALS mutations.

Also, what genotype is shown in the example gels? Does skipping of the two affected exons lead to a frameshift?

Referee #3:

the work of Fratta et al had improved in revisions and I believe it is justified to publish the current version at the EMBO journal. I do not have additional concerns.

Referee #2:

Overall the manuscript improved and now contains a thorough characterization of the two TDP-43 ENU mice.

We thank the reviewer and also referees 1 and 3 for the positive reception of our revised manuscript.

The question how relevant these effects are for ALS is not fully answered. The data from a mouse line with a known pathogenic TDP-43 mutation (Q331K) and patient fibroblasts in Fig. 7 is not very convincing. I was puzzled by the apparent discrepancy of the large effects in the quantification and the virtual absence of effects in the gel pictures (especially Fig. 7C/D - compare Fig 3C) until I noticed that the y-axis in most graphs is not starting at 0. For example, for *Pacrg1* the exon inclusion merely drops from 96% to 91%. This may be statistically significant but is not highly convincing. All y-axes in Fig 7 should start at 0 and the interpretation has to be more cautious.

- We have modified the y-axis accordingly, the data remains the same as before.
- We have toned down the interpretation of results and have added the following sentence to the end of the Results section: "...although further studies will be necessary to assess how widespread the GOF induced by ALS-causing mutations is".
- We agree that the changes in Q331K, although significant and within in the range of dPSI>5% defined in our genome-wide SE analysis, are smaller than in *LCDmut*. We have added the following sentence to address this point: "The effect observed in Q331K was significant, but smaller than in *LCDmut*, possibly due to the younger age of the analysed mice or the different genetic background".

In Fig 7E, quantification for all 7 SEs tested in the patient fibroblasts should be shown using the same presentation/calculation as in the other figures (PSI instead of normalized skiptic exons). I assume these effects will also appear much smaller then.

- We have now included data for SEs that did not show changes in patient fibroblasts in an additional figure: Appendix Figure S10.
- As stated in the Methods (see below), we have used a different measurement for the human fibroblast data because a three primer PCR was used to detect skiptic exons in this context, compared to a two primer PCR used in mouse. The amplicons generated from different primer pairs do not allow to calculate a PSI, and we have consequently normalised WT to 100% and expressed fold change.

This was stated in the Methods: "When analysing SEs in human fibroblasts, as a combination of three primers was used, we did not calculate PSI, but calculated the ratio between SE band and the exon inclusion band and expressed results after normalising the mean of healthy controls to 100%". We have now added the following sentence to Figure 7 legend to make this more clear to the resder: "ratio between SE and the exon inclusion bands, after normalising the mean of healthy controls to 100%, is used instead of PSI, as amplification was performed using a three primer PCR".

The current presentation in Fig 7 greatly exaggerates how wide-spread exon skipping occurs in bona fide ALS mutations.

- As above, we have toned down the interpretation of results and have added the following sentence to the end of the Results section: "although further studies will be necessary to assess how widespread the GOF induced by ALS-causing mutations is".

Also, what genotype is shown in the example gels?

- We have now uploaded data and gels from all genotypes in Source Data and in Appendix Figure S10. We have indicated in figure legend the genotypes represented.

Does skipping of the two affected exons lead to a frameshift?

- In one case it does (*SLC6A6*), in the other it doesn't (*PLOD1*).